# What Makes Graph Neural Networks Miscalibrated?

**Hans Hao-Hsun Hsu**[*1]    **Yuesong Shen**[*1,2]    **Christian Tomani** [1,2]    **Daniel Cremers** [1,2]

[1] Technical University of Munich, Germany
[2] Munich Center for Machine Learning, Germany
{hans.hsu, yuesong.shen, christian.tomani, cremers}@tum.de

## Abstract

Given the importance of getting calibrated predictions and reliable uncertainty estimations, various post-hoc calibration methods have been developed for neural networks on standard multi-class classification tasks. However, these methods are not well suited for calibrating graph neural networks (GNNs), which presents unique challenges such as accounting for the graph structure and the graph-induced correlations between the nodes. In this work, we conduct a systematic study on the calibration qualities of GNN node predictions. In particular, we identify five factors which influence the calibration of GNNs: general under-confident tendency, diversity of nodewise predictive distributions, distance to training nodes, relative confidence level, and neighborhood similarity. Furthermore, based on the insights from this study, we design a novel calibration method named Graph Attention Temperature Scaling (GATS), which is tailored for calibrating graph neural networks. GATS incorporates designs that address all the identified influential factors and produces nodewise temperature scaling using an attention-based architecture. GATS is accuracy-preserving, data-efficient, and expressive at the same time. Our experiments empirically verify the effectiveness of GATS, demonstrating that it can consistently achieve state-of-the-art calibration results on various graph datasets for different GNN backbones.[2]

## 1   Introduction

Graph-structured data, such as social networks, knowledge graphs and internet of things, have widespread presence and learning on graphs using neural networks has been an active area of research. For node classification on graphs, a wide range of graph neural network (GNN) models, including GCN [11], GAT [32] and GraphSAGE [7], have been proposed to achieve high classification accuracy.

This said, high accuracy is not the only desideratum for a classifier. Especially, reliable uncertainty estimation is crucial for applications like safety critical tasks and active learning. Neural networks are known to produce poorly calibrated predictions that are either overconfident or under-confident [5, 33]. To mitigate this issue a variety of post-hoc calibration methods [5, 15, 35, 30, 6] have been introduced over the last few years for calibrating neural networks on standard multi-class classification problems. However, calibration of GNNs, in the context of node classification on graphs, is currently still an underexplored topic. While it is possible to apply existing calibration methods designed for multi-class classification to GNNs in a nodewise manner, this does not address the specific challenges of node classification on graphs. Especially, node predictions in a graph are not i.i.d. but correlated, and we are tackling a structured prediction problem [20]. A uniform treatment when calibrating node predictions would fail to account for the structural information from graphs and the non i.i.d. behavior of node predictions.

---

[*]Equal contribution
[2]Source code available at `https://github.com/hans66hsu/GATS`

36th Conference on Neural Information Processing Systems (NeurIPS 2022).

**Our contribution.** In this work, we focus on calibrating GNNs for the node classification task [11, 32]. First, we aim at understanding the specific challenges posed by GNNs by conducting a systematic study on the calibration qualities of GNN node predictions. Our study reveals five factors that influence the calibration performance of GNNs: general under-confident tendency, diversity of nodewise predictive distributions, distance to training nodes, relative confidence level, and neighborhood similarity. Second, we develop Graph Attention Temperature Scaling (GATS) approach, which is designed in a way that accounts for the aforementioned influential factors. GATS generates nodewise temperatures that calibrate GNN predictions based on the graph topology. Third, we conduct a series of GNN calibration experiments and empirically verify the effectiveness of GATS in terms of calibration, data-efficiency, and expressivity.

## 2 Related work

For standard multi-class classification tasks, a variety of post-hoc calibration methods have been proposed in order to make neural networks uncertainty aware: temperature scaling (TS) [5], ensemble temperature scaling (ETS) [35], multi-class isotonic regression (IRM) [35], Dirichlet calibration [15], spline calibration [6], etc. Additionally, calibration has been formulated for regression tasks [13]. More generally, instead of transforming logits after training a classifier, a plethora of methods exists that modify either the model architecture or the training process itself. This includes methods that are based on Bayesian paradigm [10, 1, 4, 17, 34], evidential theory [25], adversarial calibration [29] and model ensembling [16]. One common caveat of these methods is the trade-off between accuracy and calibration, which oftentimes do not go hand in hand. Post-hoc methods like temperature scaling, on the other hand, are accuracy preserving. They ensure that the per node logit rankings are unaltered.

Calibration of GNNs is currently a substantially less explored topic. Nodewise post-hoc calibration on GNNs using methods developed for the multi-class setting has been empirically evaluated by Teixeira et al. [28]. They show that these methods, which perform uniform calibration of nodewise predictions, are unable to produce calibrated predictions for some harder tasks. Wang et al. [33] observe that GNNs tend to be under-confident in contrast to the majority of multi-class classifiers, which are generally overconfident [5]. Based on their findings, Wang et al. [33] propose the CaGCN approach, which attaches a GCN on top of the backbone GNN for calibration. Some approaches improve the uncertainty estimation of GNNs by adjusting model training. This includes Bayesian learning approaches [37, 8] and methods based on the evidential theory [38, 27].

## 3 Problem setup for GNN calibration

We consider the problem of calibrating GNNs for node classification tasks: given a graph $\mathcal{G} = (\mathcal{V}, \mathcal{E})$, the training data consist of nodewise input features $\{x_i\}_{i \in \mathcal{V}} \in \mathcal{X}$ and ground-truth labels $\{y_i\}_{i \in \mathcal{L}} \in \mathcal{Y} = \{1, \ldots, K\}$ for a subset $\mathcal{L} \subset \mathcal{V}$ of nodes, and the goal is to predict the labels $\{y_i\}_{i \in \mathcal{U}} \in \mathcal{Y}$ for the rest of the nodes $\mathcal{U} = \mathcal{V} \setminus \mathcal{L}$. A graph neural network tackles the problem by producing nodewise probabilistic forecasts $\hat{p}_i$. These forecasts yield the corresponding label predictions $\hat{y}_i := \arg\max_y \hat{p}_i(y)$ and confidences $\hat{c}_i := \max_y \hat{p}_i(y)$. The GNN is calibrated when its probabilistic forecasts are reliable, e.g., for predictions with confidence 0.8, they should be correct 80% of the time. Formally, a GNN is *perfectly calibrated* [33] if

$$\forall c \in [0, 1], \quad \mathbb{P}(y_i = \hat{y}_i | \hat{c}_i = c) = c. \tag{1}$$

In practice, we quantify the calibration quality with the expected calibration error (ECE) [21, 5]. We follow the commonly used definition from Guo et al. [5] which uses a equal width binning scheme to estimate calibration error for any node subset $\mathcal{N} \subset \mathcal{V}$: the predictions are regrouped according to $M$ equally spaced confidence intervals, i.e. $(B_1, \ldots, B_M)$ with $B_m = \{j \in \mathcal{N} \mid \frac{m-1}{M} < \hat{c}_j \leq \frac{m}{M}\}$, and the expected calibration error of the GNN forecasts is defined as

$$\text{ECE} = \sum_{m=1}^{M} \frac{|B_m|}{|\mathcal{N}|} \Big| \text{acc}(B_m) - \text{conf}(B_m) \Big|, \quad \text{with} \tag{2}$$

$$\text{acc}(B_m) = \frac{1}{|B_m|} \sum_{i \in B_m} \mathbf{1}(y_i = \hat{y}_i) \quad \text{and} \quad \text{conf}(B_m) = \frac{1}{|B_m|} \sum_{i \in B_m} \hat{c}_i. \tag{3}$$

# 4 Factors that influence GNN calibration

To design calibration methods adapted to GNNs, we need to figure out the particular factors that influence the calibration quality of GNN predictions. For this we train a series of GCN [11] and GAT [32] models on seven graph datasets: Cora [24], Citeseer [24], Pubmed [18], Amazon Computers [26], Amazon Photo [26], Coauthor CS [26], and Coauthor Physics [26]. We summarize the dataset statistics in Appendix A.1. Details about model training are provided in Appendix A.2 for reproducibility. To compare with the standard multi-class classification case, we additionally train ResNet-20 [9] models on the CIFAR-10 image classification task [12] as a reference.

Our experiments uncover five decisive factors that affect the calibration quality of GNNs. In the following we discuss them in detail.

## 4.1 General under-confident tendency

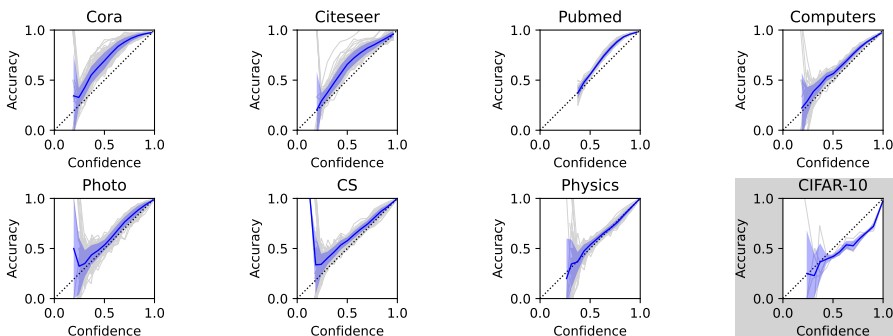

Figure 1: Reliability diagrams of GCN models trained on various graph datasets. We see a general tendency of under-confident predictions (plots above the diagonal) except the Physics dataset. This is in contrast to the overconfident behavior of multi-class image classification using CNNs (in gray).

Starting with a global perspective, we notice that GNNs tend to produce under-confident predictions. In Figure 1 we plot the reliability diagrams [19] for results on different graph datasets using GCN. Similar to Wang et al. [33], we see a general trend of under-confident predictions for GNNs. This is in contrast to the standard multi-class image classification case which has overconfident behavior. Also, it is interesting to see that this under-confident trend can be more or less pronounced depending on the dataset. For Coauthor Physics, the predictions are well calibrated and have no significant bias.

Results using GAT models lead to similar conclusions and are provided in Appendix B.1.

## 4.2 Diversity of nodewise predictive distributions

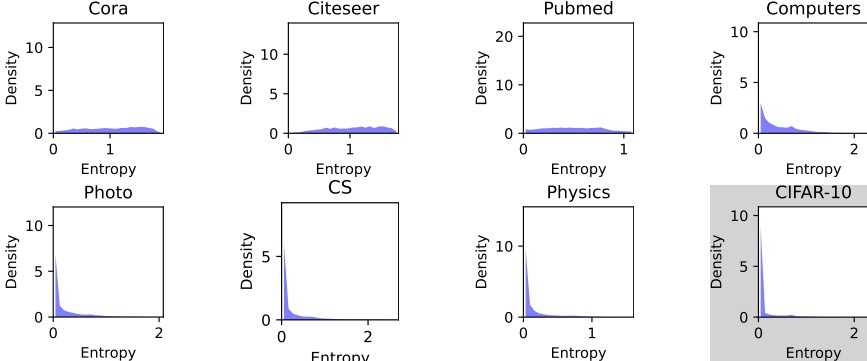

Figure 2: Entropy distributions of GCN predictions on graph datasets. Compared to the standard classification case, GNN predictions tend to be more dispersed, reflecting their disparate behaviors.

Contrary to the standard multi-class case, GNN outputs can have varying roles depending on their positions in the graph, which means that their output distributions could exhibit dissimilar behaviors. This is empirically evident in Figure 2, where we visualize the entropy distributions of GCN output predictions v.s. the standard multi-class results (GAT results are available in Appendix B.2). We see that the entropies of GNN outputs have more spread-out distributions, which indicates that they have distinct roles and behaviors in graphs.

In terms of GNN calibration, this observation implies that uniform node-agnostic adjustments like temperature scaling [5] might be insufficient for GNNs, whereas nodewise adaptive approaches could be beneficial.

### 4.3 Distance to training nodes

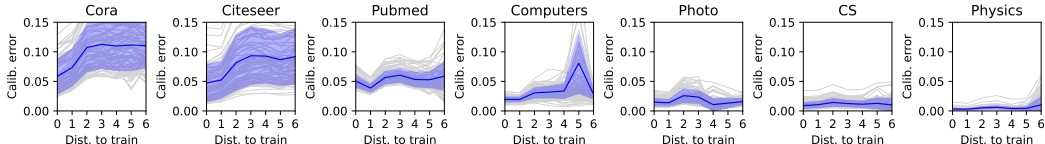

Figure 3: Nodewise calibration error of GCN results depending on the minimum distance to training nodes. We observe that training nodes and their neighbors tend to be better calibrated.

A graph provides additional structural information for its nodes. One insightful feature is the minimum distance to training nodes. We discover that nodes with shorter distances, especially the training nodes themselves and their direct neighbors, tend to be better calibrated.

To evaluate the calibration quality nodewise, we propose the *nodewise calibration error*, which is based on the binning scheme used to compute the global expected calibration error (ECE) [21, 5]: for each node, we find its corresponding bin depending on its predicted confidence, and the calibration error of this bin is assigned to be its nodewise calibration error.

Using this nodewise metric, in Figure 3 we visualize the influence of minimum distance to training nodes on the nodewise calibration quality (c.f. Appendix B.3 for GAT results). We see that nodes close to training ones typically have lower nodewise calibration error. This suggests that minimum distance to training nodes can be useful for GNN calibration.

### 4.4 Relative confidence level

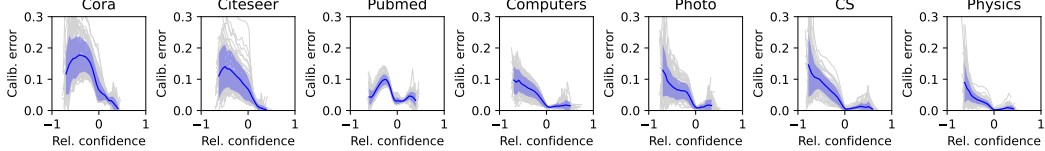

Figure 4: Nodewise calibration error of GCN results depending on the relative confidence level. We observe that nodes which are less confident than their neighbors tend to have worse calibration.

Another important structural information is the neighborhood relation. We find out that the relative confidence level $\delta \hat{c}_i$ of a node $i$, i.e., the difference between the nodewise confidence $\hat{c}_i$ and the average confidence of its neighbors

$$\delta \hat{c}_i = \hat{c}_i - \frac{1}{|n(i)|} \sum_{j \in n(i)} \hat{c}_j \qquad (4)$$

has an interesting correlation to the nodewise calibration quality. In Figure 4 we show the relation between the relative confidence level of a node and its nodewise calibration error (c.f. Appendix B.4 for GAT results). Especially, We observe that nodes which are less confident than their neighbors tend to have worse calibration, and it is in general desirable to have comparable confidence level w.r.t. the neighbors. For GNN calibration, the relative confidence level $\delta \hat{c}_i$ can be a useful node feature to consider.

## 4.5 Neighborhood similarity

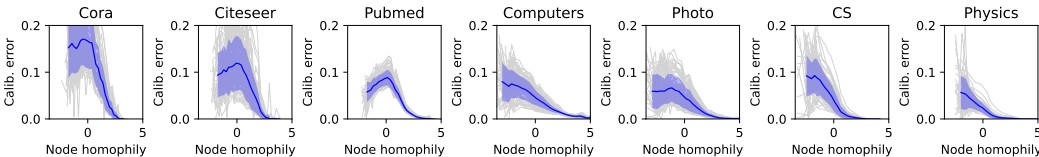

Figure 5: Nodewise calibration error of GCN results depending on the node homophily. Nodes with strongly agreeing neighbors tend to have significantly lower calibration errors.

Furthermore, we find that different neighbors tend to introduce distinct influences. For assortative graphs which are the focus of this work, we find out that calibration of nodes are affected by *node homophily*, i.e., whether a node tends to have the same label prediction as its neighbors. For a node with $n_a$ agreeing neighbors and $n_d$ disagreeing ones, we measure the node homophily as

$$\text{Node homophily} = \log\left(\frac{n_a + 1}{n_d + 1}\right), \tag{5}$$

where positive values indicate greater ratio of agree neighbors and vice versa.

Figure 5 summarizes the variation of nodewise calibration error w.r.t. the node homophily for different graph datasets (c.f. Appendix B.5 for GAT results). We find out that nodewise calibration errors tend to decrease significantly for nodes with strongly agreeing neighbors. This suggests that neighborhood predictive similarity should be considered when doing GNN calibration.

## 5 Graph attention temperature scaling (GATS)

Based on the findings in Section 4, we design a new post-hoc calibration method, named *Graph Attention Temperature Scaling (GATS)*, which is tailored for GNNs.

### 5.1 Formulation and design of GATS

To obtain a calibration method that is adapted to the graph structure $\mathcal{G} = (\mathcal{V}, \mathcal{E})$ and reflects the observed influential factors in Section 4, the graph attention temperature scaling approach extends the temperature scaling [5] method to produce a distinct temperature $T_i$ for each node $i \in \mathcal{V}$. $T_i$ is then used to scale the uncalibrated nodewise output logits $z_i$ and produce calibrated node predictions $\hat{p}_i$

$$\forall i \in \mathcal{V}, \quad \hat{p}_i = \text{softmax}\left(\frac{z_i}{T_i}\right). \tag{6}$$

**Formulation of $T_i$.** The nodewise temperature $T_i$ should address the five factors discussed in Section 4. We achieve this via the following considerations:

- We introduce a global bias parameter $T_0$ to account for the general under-confident tendency;
- To tackle the diverse behavior of node predictions, we learn a nodewise temperature contribution $\tau_i$ based on the predicted nodewise logits $z_i$;
- To incorporate the relative confidence w.r.t. neighbors, we introduce $\delta\hat{c}_i$ from Eq. 4 as an additional contribution term scaled by a learnable coefficient $\omega$;
- To model the influence of neighborhood similarity, we use an attention mechanism [31] to aggregate neighboring contributions $\tau_j$ with attention coefficients $\alpha_{i,j}$ depending on the output similarities between the neighbors $i$ and $j$;
- Distance to training nodes is used to introduce a nodewise scaling factor $\gamma_i$ to adjust the node contribution and the aggregation process. It is learnable for training nodes and their direct neighbors and fixed to 1 for the rest:

$$\gamma_i = \begin{cases} \gamma_t, & \text{if } i \text{ is a training node} \\ \gamma_n, & \text{if } i \text{ is a neighbor of training node }, \quad \gamma_t, \gamma_n \text{ learnable parameters.} \\ 1, & \text{otherwise} \end{cases} \tag{7}$$

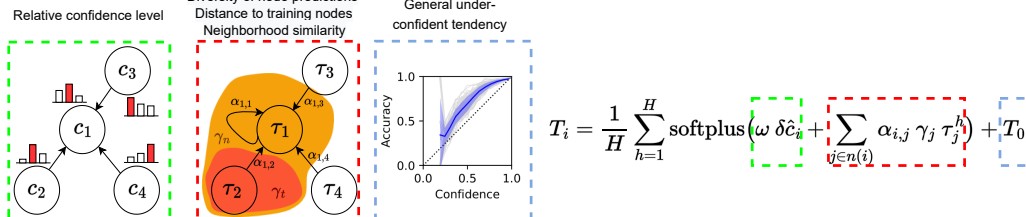

Figure 6: Illustration of graph attention temperature scaling (GATS) for a graph with four nodes where node 2 is a training node and node 1 is the target node which aggregates information from the other three nodes. GATS incorporates the five factors discussed in Section 4 and produces nodewise temperatures $T_i$ using the graph structure and an attention mechanism.

Putting together the above components, the nodewise temperature $T_i$ has the following expression:

$$\forall i \in \mathcal{V}, \quad T_i = \frac{1}{H} \sum_{h=1}^{H} \text{softplus} \left( \omega \, \delta \hat{c}_i + \sum_{j \in \hat{n}(i)} \alpha_{i,j} \, \gamma_j \, \tau_j^h \right) + T_0. \tag{8}$$

Here we have a multi-head formulation where $h$ indicates the $h$-th attention head, and $\hat{n}(i)$ denotes the neighbors of node $i$ including self-loop. We uses 8 heads ($H = 8$), which works well in practice. (c.f. Section 6.3.)

**Defining $\tau_i^h$.** Nodewise contribution $\tau_i^h$ are computed as the outputs of parameterized linear layers $\phi^h(\cdot; \theta)$ which take transformed nodewise output logits $\tilde{z}_i$ as input

$$\forall i \in \mathcal{V}, \forall h \in [1, H], \quad \tau_i^h = \phi^h(\tilde{z}_i; \theta^h) = (\theta^h)^\top \tilde{z}_i. \tag{9}$$

The transformed nodewise output logits $\tilde{z}_i$ are produced as follows: we first normalize the original logits $z_i$ to range $[0, 1]$, then sort the classwise logits for each node. This makes the linear layers $\phi^h$ focus on the general logit distributions rather than class predictions. A similar idea has been explored by Rahimi et al. [23], where they show that intra order-preserving functions improve the model calibration. Here we find out that this sorting-based transformation helps GATS to learn useful representations of nodewise contributions $\tau_i^h$. (c.f. Section 6.3.)

**Defining $\alpha_{i,j}$.** The attention coefficients $\alpha_{i,j}$ are defined based on the neighbor similarity, which is determined by the inner product between rescaled nodewise logits $z_i/\gamma_i$ and $z_j/\gamma_j$. Inspired by Veličković et al. [32], we compute the attention coefficients $\alpha_{i,j}$ as follows:

$$(\alpha_{i,j})_{j \in \hat{n}(i)} = \underset{j \in \hat{n}(i)}{\text{softmax}} \left( \text{leakyReLU} \left( \frac{1}{\gamma_i \, \gamma_j} z_i^\top z_j \right) \right). \tag{10}$$

## 5.2 Calibration properties of GATS

Zhang et al. [35] propose three desiderata for calibration methods: accuracy-preserving, data-efficient, and expressive. GATS fulfills all of them:

- GATS is accuracy-preserving: since all node predictions are scaled by inverse temperatures $1/T_i$ which are positive scalars, the order of output logits is preserved;

- GATS is data-efficient. It is a parametric calibration model with $C \cdot H + 4$ learnable parameters $(T_0, \omega, \gamma_t, \gamma_n, (\theta^h)_{1 \le h \le H})$. According to Zhang et al. [35], parametric methods are already data-efficient;

- GATS is expressive, as it produces nodewise temperatures $T_i$ adapted to the graph structure.

Experiments in Section 6.2 empirically confirm the data-efficiency and expressivity of GATS.

### 5.3 Comparison with CaGCN

It is interesting to compare our proposed GATS approach to the CaGCN method proposed by Wang et al. [33], since both approaches aim at calibrating GNNs, and both make use of the graph structure to produce nodewise temperatures. While CaGCN uses a GCN to generate nodewise temperatures straightforwardly, GATS uses an attention mechanism which differentiates the influence from various neighbors. GATS also integrates a series of careful designs following the insights from the study in Section 4. Experiments in Section 6.1 shows that GATS tends to produce better calibration results compared to CaGCN.

## 6 Experiments

To evaluate the performance of GATS on GNN calibration and understand the effects of its designs, we conduct a series of experiments for baseline comparison and ablation study. We use two representative GNNs: GCN and GAT, which are trained on the seven aforementioned graph datasets plus a larger graph, CoraFull [2], for post-hoc calibration. We display the ECE results with $M = 15$ bins. We use the expected calibration error (ECE) [21, 5] with $M = 15$ bins as an evaluation metric, and follow an experimental protocol similar to Kull et al. [15, 14]: For all the experiments, we randomly split the labeled/unlabeled ($15\%/85\%$) data five times, and use three-fold internal cross-validation of the labeled data to train the GNNs and the calibrators. We also utilize five random initializations, resulting in 75 total runs for each experiment. We provide detailed experimental settings in Appendix A.

### 6.1 Performance comparison

We benchmark GATS against existing baselines on a variety of GNN calibration tasks. We compare GATS with the following baselines:

- **Temperature scaling (TS)** [5] simply uses a global temperature to scale the logits.
- **Vector scaling (VS)** [5] scales the logits separately over the class dimension and additionally introduces a classwise bias for the recalibrated output logits.
- **Ensemble temperature scaling (ETS)** [35] learns a mixture of uncalibrated, TS-calibrated, and uniform probabilistic outputs.
- **GCN as a calibration function (CaGCN)** [33] is specifically designed for calibrating GNNs. It uses a GCN on top to generate nodewise temperatures.

Additionally, we also report the ECEs of uncalibrated predictions as a reference. Among the above baselines, TS, VS, and ETS are calibration methods designed for standard classification cases and operate on nodes uniformly. CaGCN on the other hand performs separate nodewise adjustments and uses the graph structure, similar to our proposed GATS approach.

For the post-hoc calibration experiments, we fix the weight of the trained GNN backbones and adjust the parameters of the calibration methods on the validation set. Negative log-likelihood is chosen as the objective for the calibration process. We provide details of method configurations and calibration settings in Appendix A.3. Table 1 summarizes the calibration results.

Overall, we observe that GATS consistently produces well-calibrated predictions for all graph datasets and GNN backbones. Except for the GAT model trained on Pubmed (3rd best) and the GCN model trained on Amazon Computers (2nd best), GATS achieves the highest calibration quality in all cases.

Also, it is interesting to see that for all cases the best result is achieved by methods which use the graph structure and produce adapted adjustments for different nodes. This demonstrates the necessity of designing calibration methods that address the unique challenges posed by GNN calibration.

Although CaGCN can get the best results for Pubmed using GAT and Amazon Computers using GCN, we see that its performance is rather unstable for different scenarios, and sometimes it can even produce worse calibration results than the uncalibrated baseline. Using their proposed margin-based loss did not help in our settings. We suspect that CaGCN might have an overly complex architecture for the task, and it cannot differentiate neighborhood influences with the common normalized adjacency matrix. Our proposed GATS model does not have this issue. It has consistent and good calibration performance in all cases.

Table 1: GNN calibration results in terms of ECE (in percentage, lower is better) of GATS and other baseline methods on various graph datasets. Overall, GATS achieves state-of-the-art performance, getting the best results in most scenarios. Also, all best results are achieved by methods that consider the graph structure. This shows the need for dedicated methods to tackle GNN calibration.

| Dataset | Model | Uncal | TS | VS | ETS | CaGCN | GATS |
|---|---|---|---|---|---|---|---|
| Cora | GCN | 13.04±5.22 | 3.92±1.29 | 4.36±1.34 | 3.79±1.35 | 5.29±1.47 | **3.64±1.34** |
| | GAT | 23.31±1.81 | 3.69±0.90 | 3.30±1.12 | 3.54±1.01 | 4.09±1.06 | **3.18±0.90** |
| Citeseer | GCN | 10.66±5.92 | 5.15±1.50 | 4.92±1.44 | 4.65±1.69 | 6.86±1.41 | **4.43±1.30** |
| | GAT | 22.88±3.53 | 4.74±1.47 | 4.25±1.48 | 4.11±1.64 | 5.75±1.31 | **3.86±1.56** |
| Pubmed | GCN | 7.18±1.51 | 1.26±0.28 | 1.46±0.29 | 1.24±0.30 | 1.09±0.52 | **0.98±0.30** |
| | GAT | 12.32±0.80 | 1.19±0.36 | 1.00±0.32 | 1.20±0.32 | **0.98±0.31** | 1.03±0.32 |
| Computers | GCN | 3.00±0.80 | 2.65±0.57 | 2.70±0.63 | 2.58±0.70 | **1.72±0.53** | 2.23±0.49 |
| | GAT | 1.88±0.82 | 1.63±0.46 | 1.67±0.52 | 1.54±0.67 | 2.03±0.80 | **1.39±0.39** |
| Photo | GCN | 2.24±1.03 | 1.68±0.63 | 1.75±0.63 | 1.68±0.89 | 1.99±0.56 | **1.51±0.52** |
| | GAT | 2.02±1.11 | 1.61±0.63 | 1.63±0.69 | 1.67±0.73 | 2.10±0.78 | **1.48±0.61** |
| CS | GCN | 1.65±0.92 | 0.98±0.27 | 0.96±0.30 | 0.94±0.24 | 2.27±1.07 | **0.88±0.30** |
| | GAT | 1.40±1.25 | 0.93±0.34 | 0.87±0.35 | 0.88±0.33 | 2.52±1.04 | **0.81±0.30** |
| Physics | GCN | 0.52±0.29 | 0.51±0.19 | 0.48±0.16 | 0.52±0.19 | 0.94±0.51 | **0.46±0.16** |
| | GAT | 0.45±0.21 | 0.50±0.21 | 0.52±0.20 | 0.50±0.21 | 1.17±0.42 | **0.42±0.14** |
| CoraFull | GCN | 6.50±1.26 | 5.54±0.43 | 5.76±0.42 | 5.38±0.49 | 5.86±2.52 | **3.76±0.74** |
| | GAT | 4.73±1.39 | 4.00±0.50 | 4.17±0.43 | 3.89±0.56 | 6.55±3.69 | **3.54±0.63** |

We also observe that the results tend to have high variations, since GNN backbones tend to predict highly varying results when trained with different initial weights and random splits [26]. We ensure the reliability of the results by averaging over a total of 75 runs with various initial weights and random splits for each case.

## 6.2 Data-efficiency and expressivity of GATS

Furthermore, we analyze the data-efficiency and the expressivity of GATS for GNN calibration. For this we reuse the GNN models trained on the CoraFull dataset, and consider the influence of calibration sample size on the GNN calibration performance. For comparison we also report the corresponding results using ensemble temperature scaling and CaGCN. Figure 7 visualizes the results with GCN backbone. The results for GAT backbone are summarized in Appendix D.

Overall, we see that GATS is both data-efficient and expressive. It requires few calibration samples to get decent calibration performance. This is in contrast to CaGCN which needs more than 5% of nodes for calibration to get acceptable results. Compared to ETS, GATS is more expressive and has a considerably lower calibration error for CoraFull, which is a large graph dataset.

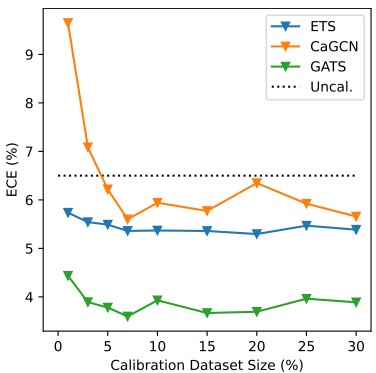

Figure 7: ECEs (in percentage) on CoraFull for ETS, CaGCN, and GATS using various amounts of calibration data. We see that GATS is data-efficient and expressive for GNN calibration.

## 6.3 Ablation study

To empirically analyze the effect of various GATS design choices, we conduct a series of ablation study experiments in this section. Overall, we notice that all designs are advantageous and removing any of them leads to a general decrease in performance.

Table 2: Ablation study results in terms of ECE (in percentage) for various GATS designs. Overall, all designs are beneficial and removing any of them leads to worse results in general.

| Dataset | Model | w/o $T_0$ | w/o $\gamma_i$ | w/o $\delta\hat{c}_i$ | w/o attention | w/o sorting | GATS |
|---|---|---|---|---|---|---|---|
| Cora | GCN | 3.72±1.20 | 3.80±1.51 | 3.71±1.18 | 3.63±1.48 | 4.35±1.77 | 3.64±1.34 |
| | GAT | 3.25±1.00 | 3.46±1.00 | 3.24±0.89 | 3.69±0.96 | 4.18±1.70 | 3.18±0.90 |
| Citeseer | GCN | 5.50±1.76 | 4.73±1.45 | 4.49±1.30 | 4.95±1.56 | 5.87±1.99 | 4.43±1.30 |
| | GAT | 3.56±1.73 | 4.39±1.46 | 3.87±1.55 | 4.81±1.54 | 4.99±2.34 | 3.86±1.56 |
| Photo | GCN | 2.20±0.88 | 1.60±0.64 | 1.54±0.52 | 1.59±0.67 | 1.68±0.61 | 1.51±0.52 |
| | GAT | 2.37±1.01 | 1.53±0.63 | 1.47±0.62 | 1.62±0.66 | 1.77±0.70 | 1.48±0.61 |

**Effect of global bias $T_0$.**    We consider the effect of global bias $T_0$ by comparing to a GATS variant without it (i.e., setting $T_0 = 0$ in Eq. (8)). Its results are collected in column "w/o $T_0$" of Table 2. Overall we see that a learnable bias $T_0$ is beneficial in most cases.

**Effect of nodewise scaling factor $\gamma_i$.**    The nodewise scaling factors $\gamma_i$ provide custom adjustment for training nodes and their neighbors. Removing its influence can be done by fixing all $\gamma_i$ to one in Eqs. (8) and (10). The results of the variant without $\gamma_i$ are recorded in column "w/o $\gamma_i$" of Table 2. They are worse in general, suggesting that nodewise scaling factors $\gamma_i$ are indeed helpful.

**Effect of relative confidence level $\delta\hat{c}_i$.**    To evaluate the impact of introducing nodewise relative confidence level $\delta\hat{c}_i$, we create a GATS variant where $\omega$ in Eq. (8) is fixed to zero, which effectively removes the influence of the relative confidence level $\delta\hat{c}_i$. In column "w/o $\delta\hat{c}_i$" of Table 2 we have the results corresponding to this variant, which is in general slightly worse than the standard GATS which includes $\delta\hat{c}_i$.

**Effect of attention-based aggregation.**    To understand the role played by the attention mechanism, we create a GATS variant which completely removes the attention related term ($\sum_{j \in \hat{n}(i)} \alpha_{i,j} \gamma_j \tau_j^h$) from Eq. (8). Its performance is shown in column "w/o attention" of Table 2. Again, we observe that removing the attention-based component tends to worsen the calibration performance.

**Effect of logit sorting.**    GATS uses normalized and sorted logits $\tilde{z}_i$ as input to generate nodewise temperature contributions $\tau_i^h$. And we find that this sorting transform is essential for learning good representations of $\tau_i^h$. In column "w/o sorting" of Table 2 we have the results for GATS variants which uses the logits without sorting to compute $\tau_i^h$. And we observe that this deteriorates the calibration performance.

Table 3: Calibration results (ECE in percentage) of GATS models with different numbers of attention heads. 8 attention heads are sufficient for optimal GNN calibration results.

| Dataset | Model | Number of Heads | | | | |
| | | 1 | 2 | 4 | 8 | 16 |
|---|---|---|---|---|---|---|
| Cora | GCN | 3.95±1.45 | 3.75±1.42 | 3.79±1.45 | 3.66±1.33 | 3.50±1.25 |
| | GAT | 3.48±1.22 | 3.49±1.18 | 3.43±1.15 | 3.20±0.90 | 3.31±0.74 |
| Citeseer | GCN | 4.74±1.43 | 4.77±1.66 | 4.70±1.52 | 4.43±1.30 | 4.42±1.04 |
| | GAT | 4.53±1.66 | 4.57±1.55 | 4.20±1.50 | 3.86±1.56 | 4.20±1.54 |
| Photo | GCN | 1.51±0.54 | 1.54±0.59 | 1.52±0.58 | 1.51±0.52 | 1.50±0.51 |
| | GAT | 1.52±0.61 | 1.52±0.65 | 1.53±0.60 | 1.48±0.61 | 1.50±0.60 |

**Effect of attention head count $H$.**    Finally, we analyze the influence of multi-head count $H$ on the GNN calibration results. For this we run a series of experiments using GATS models with 1, 2, 4, 8, and 16 attention heads. The results are collected in Table 3. We see that for GCN backbones, GATS models with more attention heads tend to get better results. However, for GAT backbones, using 16 heads results in worse performance compared to 8 heads. Accounting also for the fact that doubling

the attention head count effectively doubles the computational requirements, 8 attention heads is a decent general setting for GATS.

# 7 Conclusion

In this work, we tackle the GNN calibration problem. We conduct a systematic study to analyze the calibration properties of GNNs predictions. Our study reveals five influential factors and manifests the unique challenges raised by GNN calibration. Based on the insights from our studies, we propose a novel calibrator, GATS, which accounts for the identified factors and is tailored for calibrating GNNs. GATS is accuracy-preserving, data-efficient, and expressive at the same time. Our experiments demonstrate that GATS achieves state-of-the-art performance for GNN calibration on various graph datasets and for different GNN backbones.

Our work focuses on the node classification tasks for assortative graphs, where neighbors tend to agree with each other. It is thus important to realize that the validity of the conclusions from Section 4 is limited to the assortative case, and might no longer hold for disassortative graphs [39, 22]. It can be an interesting future work to conduct similar studies for GNN calibration in the heterophilous case, especially when more established GNN architectures are available. More generally, devising calibration methodologies for other graph learning tasks such as link prediction [36] and graph classification [3] could also be an interesting direction for future research.

## Acknowledgments and Disclosure of Funding

This work was supported by the Munich Center for Machine Learning (MCML) and by the ERC Advanced Grant SIMULACRON. The authors would like to thank Nikita Araslanov for proofreading and helpful discussions, as well as the anonymous reviewers for their constructive feedback.

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
