# Appendix for
# "What makes graph neural networks miscalibrated?"

**Hans Hao-Hsun Hsu**[*][1]    **Yuesong Shen**[*][1,2]    **Christian Tomani** [1,2]    **Daniel Cremers** [1,2]

[1] Technical University of Munich, Germany
[2] Munich Center for Machine Learning, Germany
{hans.hsu, yuesong.shen, christian.tomani, cremers}@tum.de

## A   Experimental settings

### A.1   Dataset statistics

We consider eight real-world graph datasets including citation networks Cora [19], Citeseer [19], Pubmed [12], Coauthor CS [20], Coauthor Physics [20], and CoraFull [1] together with Amazon co-purchase networks Computers [20] and Photo [20]. Table 1 summarizes their statistics.

Table 1: Benchmark dataset statistics.

|  | Cora | Citeseer | Pubmed | Computers | Photo | CS | Physics | CoraFull |
|---|---|---|---|---|---|---|---|---|
| Nodes | 2,708 | 3,327 | 19,717 | 13,752 | 7,650 | 18,333 | 34,493 | 19,793 |
| Edges | 10,556 | 9,104 | 88,648 | 491,722 | 238,162 | 163,788 | 495,924 | 126,842 |
| Features | 1,433 | 3,703 | 500 | 767 | 745 | 6,805 | 8,415 | 8,710 |
| Classes | 7 | 6 | 3 | 10 | 8 | 15 | 5 | 70 |
| Homophily | 82.52% | 70.62% | 79.24% | 78.53% | 83.65% | 83.20% | 91.53% | 58.61% |

We report the homophily index proposed by Pei et al. [16], which provides a global view of the neighborhood similarity for a graph. Given a graph $\mathcal{G} = (\mathcal{V}, \mathcal{E})$, the homophily is defined as

$$\mathcal{H}(\mathcal{G}) = \frac{1}{|\mathcal{V}|} \sum_{i \in \mathcal{V}} \frac{\text{Number of node } i\text{'s neighbors who have the same label as } i}{\text{Number of } i\text{'s neighbors}}. \quad (1)$$

### A.2   Details of model training setup

We follow the setting of Shchur et al. [20] to define GCN [7] and GAT [22] models. Both models consist of 2 layers and the hidden dimension is fixed to 64. For the multi-head layer in GAT, the number of attention heads is fixed to 8 with 8 hidden units per head. We implement the models and training pipelines in PyTorch [15] and PyTorch Geometric [3]. All models are trained for a maximum of 2000 epochs, using early stopping with a patience of 100 epochs. We choose Adam [6] as the optimizer with initial learning rate 0.01. We add a weight decay of 5e-4 for Cora, Citeseer, and Pubmed, and 0 for the rest.

We use stratified sampling to randomly select 15% of the nodes as observed set, mask out the output labels of the rest 85% of the nodes for test prediction, and ensure that the nodes with the same label are split proportionally. Following Kull et al. [10, 9], we further divide the labeled set with three-fold cross-validation. The bigger portions (10%) are used as training sets and the rest (5%) are used as validation sets. The GNN models (GCN and GAT) are trained on the training set, then used to predict the masked-out test set. Figure 1 illustrates the aforementioned data partition in our experiments. In

---

[*]Equal contribution

36th Conference on Neural Information Processing Systems (NeurIPS 2022).

total, we use 5 random data splits, three-fold cross-validation for each split, and 5 random model initializations per data partition, resulting in 75 total runs for each experiment.

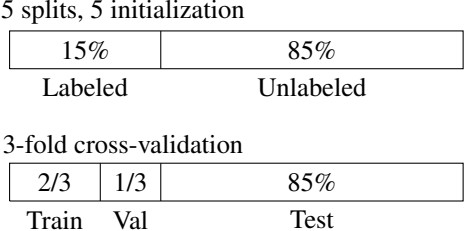

Figure 1: Data partition schema for graph datasets.

## A.3  Details of model calibration setup

We compare GATS with four scaling-based calibrators: temperature scaling (TS) [4], vector scaling (VS) [4], ensemble temperature scaling (ETS) [24] , CaGCN [23]. The calibrators are trained on the validation set using the negative log-likelihood (NLL) loss and validated on the training set for early stopping and hyperparameter search. The optimizer configuration and the training schedule are the same as Section A.2. We observe TS and VS using Adam with weight decay 0 achieves better performance than using L-BFGS [11] in the original implementation[2]. For ETS, we follow the official implementation that uses Sequential Least SQuares Programming (SLSQP) [8]. For CaGCN, we use a two-layer GCN with 16 hidden units and choose the hyperparameters following the original paper. For GATS, we utilize one message passing layer and initialize $T_0$, $\gamma_t$, and $\gamma_n$ to 1 and $\omega$ to 0. We find the best hyperparameter using cross-validation. Table 2 shows the search space for GATS hyperparameters.

Table 2: Hyperparameter search space for GATS.

| Hyperparameter | Search space |
|---|---|
| Weight decay | 0, 1e-3, 5e-3, 1e-2, 5e-2, 1e-1, 2e-1, 3e-1 |
| Initial $T_0$ | 1, 1.5 |

## B  Additional plots

Here we include additional plots which shows the corresponding factors influencing the calibration of GAT models (c.f. Section 4). Overall, we reach the same conclusions as the GCN case.

### B.1  General under-confident tendency for GAT

Figure 2 summarizes the GAT results. We see a general tendency of under-confident predictions (plots above the diagonal) except for the Physics dataset, which differs from the overconfident behavior of multiclass image classification using CNNs.

---

[2]https://github.com/gpleiss/temperature_scaling

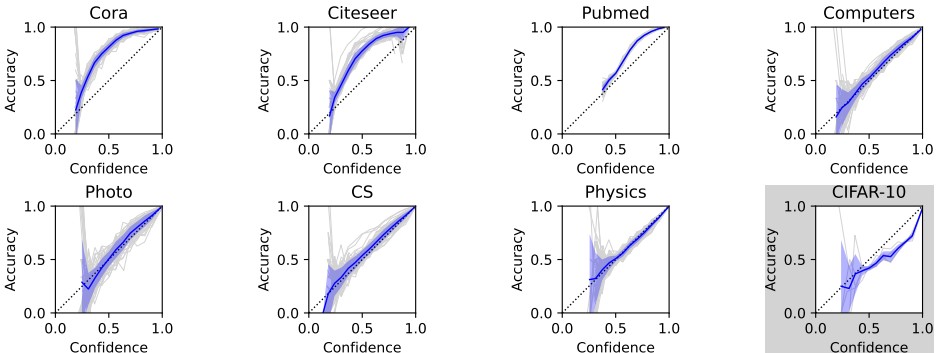

Figure 2: Reliability diagrams of GAT models trained on various graph datasets.

## B.2 Diversity of node distributions for GAT

Figure 3 shows the GAT results. Compared to the standard classification case, predictions of GAT also tend to be more spread out.

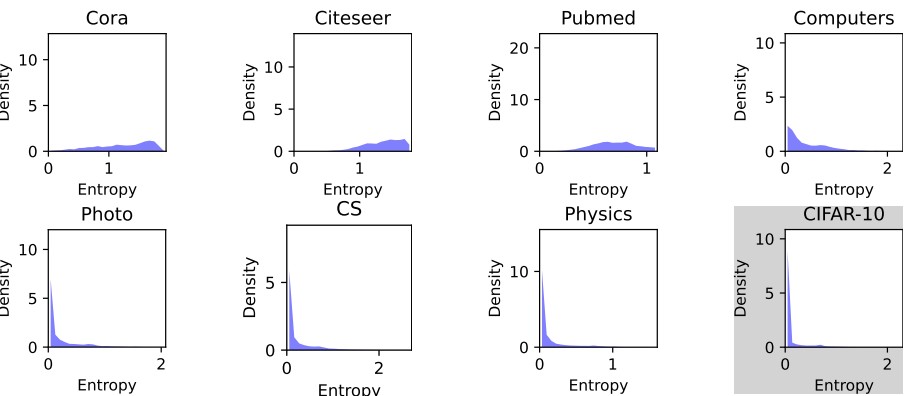

Figure 3: Entropy distributions of GAT predictions on graph datasets.

## B.3 Effect of distance to training nodes for GAT

GAT result are shown in Figure 4. We also see that training nodes and their neighbors tend to be better calibrated.

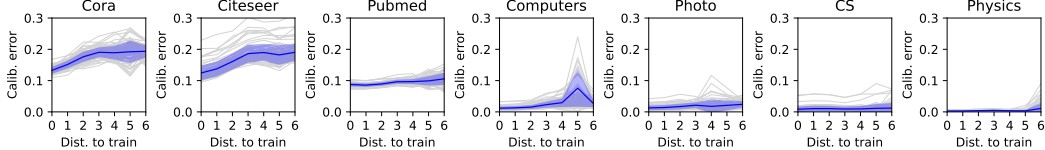

Figure 4: Nodewise calibration error of GAT results w.r.t the minimum distance to training nodes.

## B.4 Relative confidence level for GAT

The plots for the GAT case are shown in Figure 5. Similar to the GCN case, we observe that nodes which are less confident than their neighbors tend to be poorly calibrated and it is in general desirable to have a comparable confidence level w.r.t. the neighbors.

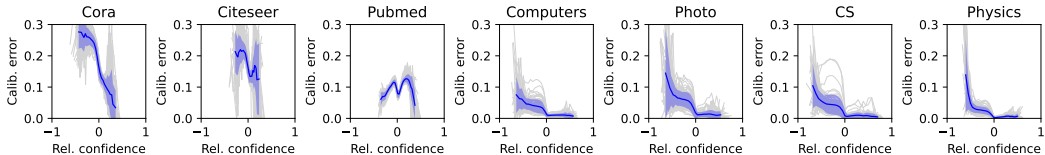

Figure 5: Nodewise calibration error of GAT results depending on the relative confidence level.

## B.5 Neighborhood similarity for GAT

Figure 6 shows tha GAT results. Analogue to the GCN case, nodes with strongly agreeing neighbors tend to have lower calibration errors.

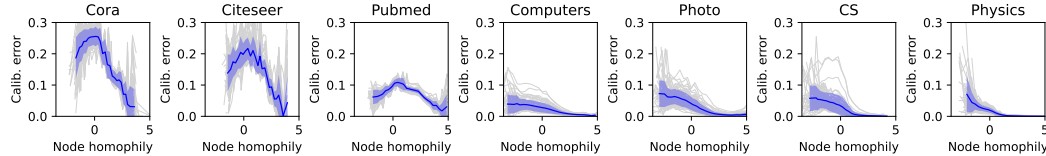

Figure 6: Nodewise calibration error of GCN results depending on the node homophily.

## C Additional calibration results

This section includes supplementary results with additional metrics and calibration methods.

### C.1 Results using additional metrics

**Variants of ECE**  Even though ECE [14, 4] is the most commonly used metric for measuring calibration, it has some limitations: (1) ECE only considers top-1 probabilistic output but can not reflect classwise calibration. (2) The binning-based estimator [14] is dependent on the choice of binning scheme. To alleviate these disadvantages, we evaluate the trained calibrators using the classwise-ECE [10] and the kernel density estimation (KDE) based KDE-ECE [24] which is a binning-free metric:

- **Classwise-ECE** measures the gap between the classwise average prediction $\text{conf}(B_{m,k})$ and the actual frequency of that class $\text{freq}(B_{m,k})$ in $M$ equally spaced bins across all classes $k \in K$:

$$\text{Classwise-ECE} = \frac{1}{K} \sum_{k}^{K} \sum_{m}^{M} \frac{|B_{m,k}|}{|\mathcal{N}|} |\text{freq}(B_{m,k}) - \text{conf}(B_{m,k})| \tag{2}$$

$$\text{freq}(B_{m,k}) = \frac{1}{|B_{m,k}|} \sum_{i \in B_{m,k}} \mathbf{1}(y_i = k) \tag{3}$$

$$\text{conf}(B_{m,k}) = \frac{1}{|B_{m,k}|} \sum_{i \in B_{m,k}} \hat{p}_{i,k} \tag{4}$$

  where $\hat{p}_{i,k}$ denotes the probability of predicting class $k$ for sample $i$ in bin $B_{m,k}$. We compute classwise-ECE using $M = 15$ bins in our implementation.

- Instead of using a binning-based estimator, **KDE-ECE** uses a kernel function $K_h$ to estimate the accuracy $\tilde{\pi}(c)$ given confidence prediction $c$ and the marginal density function $f(c)$ of

the predictive confidence:

$$\text{KDE-ECE} = \int |\tilde{\pi}(c) - c|\,\tilde{f}(c)dc \tag{5}$$

$$\tilde{\pi}(c) = \frac{\sum\limits_{i \in \mathcal{N}} \mathbf{1}(y_i = \hat{y}_i)K_h(c - \hat{c}_i)}{\sum\limits_{i \in \mathcal{N}} K_h(c - \hat{c}_i)} \tag{6}$$

$$\tilde{f}(c) = \frac{h^{-1}}{|\mathcal{N}|} \sum_{i \in \mathcal{N}} K_h(c - \hat{c}_i) \tag{7}$$

where $i \in \mathcal{N} \subset \mathcal{V}$ denotes the evaluated node, and $h$ is the bandwidth of the kernel function. We follow the official implementation[3] of KDE-ECE, where the the Triweight Kernel $K_h(u) = (1/h)\frac{35}{32}(1 - (u/h)^2)^3$ [2] on $[-1, 1]$ is chosen as the kernel function and bandwidth is calculated as $h = 1.06\sigma|\mathcal{N}|^{-1/5}$ [18] with $\sigma$ being the standard deviation of the confidence.

The classwise-ECEs are summarized in Table 3, and the KDE-ECEs are collected in Table 4. In general, we observe similar conclusions as in the confidence ECE case (c.f. Section 6.1): Overall GATS achieves the state-of-the-art calibration results.

Table 3: GNN calibration results in terms of classwise-ECE (in percentage, lower is better).

| Dataset | Model | Uncal | TS | VS | ETS | CaGCN | GATS |
|---|---|---|---|---|---|---|---|
| Cora | GCN | 4.34±1.41 | 2.06±0.27 | 2.11±0.30 | 2.07±0.26 | 2.23±0.29 | **2.03±0.24** |
|  | GAT | 7.24±0.46 | 2.35±0.23 | **2.03±0.23** | 2.34±0.24 | 2.24±0.26 | 2.34±0.28 |
| Citeseer | GCN | 4.51±1.86 | 2.85±0.40 | 2.77±0.39 | 2.82±0.42 | 3.16±0.47 | **2.74±0.39** |
|  | GAT | 8.34±1.02 | 3.12±0.52 | **2.86±0.48** | 3.09±0.51 | 3.22±0.53 | 3.10±0.58 |
| Pubmed | GCN | 4.96±0.99 | 1.38±0.26 | 1.53±0.30 | 1.39±0.26 | 1.35±0.33 | **1.26±0.26** |
|  | GAT | 8.54±0.49 | 1.94±0.31 | 1.96±0.27 | 1.94±0.31 | **1.89±0.37** | 2.00±0.32 |
| Computers | GCN | 0.97±0.16 | 0.93±0.11 | 0.91±0.12 | 0.95±0.11 | **0.84±0.10** | 0.89±0.08 |
|  | GAT | 0.83±0.13 | 0.81±0.09 | **0.80±0.10** | 0.82±0.10 | 0.84±0.11 | **0.80±0.08** |
| Photo | GCN | 0.89±0.22 | 0.78±0.12 | 0.81±0.15 | 0.78±0.15 | 0.80±0.08 | **0.76±0.10** |
|  | GAT | 0.92±0.26 | 0.84±0.15 | **0.82±0.15** | 0.84±0.17 | 0.89±0.12 | 0.83±0.15 |
| CS | GCN | 0.39±0.11 | **0.29±0.02** | 0.32±0.03 | **0.29±0.02** | 0.42±0.10 | **0.29±0.03** |
|  | GAT | 0.39±0.15 | 0.34±0.04 | 0.34±0.03 | 0.34±0.04 | 0.47±0.09 | **0.33±0.04** |
| Physics | GCN | 0.39±0.11 | 0.36±0.06 | **0.35±0.05** | 0.36±0.06 | 0.47±0.15 | 0.36±0.05 |
|  | GAT | 0.39±0.06 | 0.39±0.05 | **0.37±0.04** | 0.39±0.05 | 0.55±0.14 | 0.39±0.06 |
| CoraFull | GCN | 0.35±0.03 | 0.34±0.01 | 0.34±0.01 | 0.34±0.02 | 0.34±0.04 | **0.33±0.02** |
|  | GAT | 0.33±0.03 | 0.32±0.01 | 0.32±0.02 | 0.32±0.01 | 0.34±0.06 | **0.31±0.02** |

---

[3] https://github.com/zhang64-llnl/Mix-n-Match-Calibration

Table 4: GNN calibration results in terms of KDE-ECE (in percentage, lower is better).

| Dataset | Model | Uncal | TS | VS | ETS | CaGCN | GATS |
|---------|-------|-------|-----|-----|-----|-------|------|
| Cora | GCN | 13.35±5.07 | 3.21±1.18 | 3.56±1.25 | 3.26±1.23 | 4.17±1.48 | **3.11±1.19** |
| | GAT | 23.33±1.79 | 3.00±0.80 | **2.79±0.81** | 2.97±0.85 | 3.17±0.87 | 2.93±0.95 |
| Citeseer | GCN | 10.72±5.86 | 4.80±1.40 | 4.56±1.45 | 4.66±1.46 | 6.09±1.30 | **4.17±1.31** |
| | GAT | 22.86±3.54 | 4.42±1.46 | 3.79±1.53 | 4.32±1.42 | 5.17±1.23 | **3.66±1.58** |
| Pubmed | GCN | 7.33±1.48 | 1.32±0.27 | 1.57±0.38 | 1.35±0.29 | 1.29±0.48 | **1.07±0.26** |
| | GAT | 12.32±0.80 | 1.20±0.29 | 1.12±0.29 | 1.20±0.29 | **1.08±0.30** | 1.16±0.30 |
| Computers | GCN | 3.05±0.97 | 2.60±0.71 | 2.68±0.75 | 2.73±0.76 | **1.65±0.62** | 2.16±0.61 |
| | GAT | 1.89±1.00 | 1.62±0.63 | 1.70±0.69 | 1.67±0.72 | 1.75±0.62 | **1.47±0.52** |
| Photo | GCN | 2.59±1.29 | 1.82±0.87 | 1.94±0.92 | 1.90±0.95 | **1.65±0.45** | 1.67±0.70 |
| | GAT | 2.25±1.16 | 1.74±0.67 | 1.81±0.75 | 1.81±0.74 | 1.75±0.60 | **1.73±0.65** |
| CS | GCN | 2.14±0.98 | 1.10±0.11 | 1.11±0.17 | 1.09±0.11 | 1.95±0.90 | **1.06±0.12** |
| | GAT | 1.74±1.30 | 1.12±0.25 | 1.10±0.26 | 1.14±0.26 | 2.10±0.88 | **1.07±0.21** |
| Physics | GCN | 0.94±0.27 | 0.83±0.09 | **0.82±0.07** | 0.83±0.09 | 0.96±0.24 | 0.85±0.09 |
| | GAT | 0.84±0.10 | 0.84±0.08 | 0.85±0.09 | 0.84±0.08 | 1.07±0.24 | **0.83±0.08** |
| CoraFull | GCN | 6.46±1.30 | 5.45±0.43 | 5.66±0.41 | 5.43±0.45 | 5.73±2.54 | **3.78±0.90** |
| | GAT | 4.76±1.44 | 3.98±0.51 | 4.14±0.45 | 3.96±0.53 | 5.90±3.11 | **3.56±0.66** |

**Non calibration metrics**  Although not calibration metrics, we also report the results in terms of negative log-likelihood (Table 5) and Brier score (Table 6) for reference.

Table 5: GNN calibration results in terms of negative log-likelihood ($\times 10^{-2}$).

| Dataset | Model | Uncal | TS | VS | ETS | CaGCN | GATS |
|---------|-------|-------|-----|-----|-----|-------|------|
| Cora | GCN | 62.90±5.68 | 56.37±3.12 | 57.66±4.37 | 56.01±3.00 | 66.88±7.78 | 55.91±3.17 |
| | GAT | 75.67±2.37 | 57.51±2.87 | 55.91±4.02 | 57.15±2.70 | 60.79±4.66 | 57.02±2.33 |
| Citeseer | GCN | 90.13±6.00 | 86.99±2.74 | 87.01±2.33 | 86.61±2.56 | 92.95±4.88 | 86.18±2.35 |
| | GAT | 100.90±6.33 | 86.57±3.30 | 86.01±1.87 | 86.18±2.79 | 89.07±3.40 | 86.20±3.22 |
| Pubmed | GCN | 39.31±1.47 | 36.75±0.68 | 36.79±0.69 | 36.53±0.67 | 35.97±1.16 | 36.39±0.62 |
| | GAT | 46.87±0.98 | 40.06±0.76 | 40.06±0.74 | 40.07±0.75 | 39.78±0.77 | 40.05±0.75 |
| Computers | GCN | 42.96±1.21 | 42.93±1.17 | 42.87±1.15 | 41.08±1.31 | 43.31±3.47 | 42.49±1.25 |
| | GAT | 37.26±1.53 | 37.18±1.48 | 37.07±1.38 | 36.54±1.60 | 40.38±4.06 | 37.11±1.54 |
| Photo | GCN | 28.92±1.20 | 29.02±1.18 | 29.25±1.31 | 27.19±1.24 | 37.59±7.72 | 28.81±1.23 |
| | GAT | 26.83±1.78 | 26.82±1.61 | 26.79±1.61 | 26.40±1.76 | 32.75±5.37 | 26.93±1.77 |
| CS | GCN | 21.85±0.74 | 21.38±0.48 | 21.65±0.45 | 21.36±0.46 | 27.38±5.57 | 21.28±0.49 |
| | GAT | 24.76±1.46 | 24.57±0.87 | 24.59±0.72 | 24.49±0.85 | 29.79±3.86 | 24.49±0.83 |
| Physics | GCN | 11.95±0.41 | 11.88±0.34 | 11.90±0.33 | 11.89±0.34 | 13.00±1.27 | 11.87±0.32 |
| | GAT | 12.88±0.41 | 12.88±0.39 | 12.84±0.38 | 12.88±0.39 | 13.52±0.67 | 12.87±0.39 |
| CoraFull | GCN | 143.07±2.02 | 142.71±1.80 | 142.85±1.98 | 141.74±1.61 | 146.55±12.81 | 140.10±1.92 |
| | GAT | 139.77±2.16 | 139.57±1.89 | 139.72±1.91 | 138.97±1.86 | 150.70±17.93 | 139.06±1.84 |

### C.2  Results for additional baselines

While in the main paper we focus on "temperature scaling style" methods which directly rescale the output logits, here we compare with the following additional calibration methods which have different principles. These methods are all designed for multi-class classification and do not consider the structural information of the graph.

- **Multi-class isotonic regression (IRM)** [24] is a multi-class generalization of the non-parametric isotonic regression method;

Table 6: GNN calibration results in terms of Brier score ($\times 10^{-2}$).

| Dataset | Model | Uncal | TS | VS | ETS | CaGCN | GATS |
|---|---|---|---|---|---|---|---|
| Cora | GCN | 28.68±2.54 | 25.62±0.98 | 25.65±1.05 | 25.62±0.97 | 26.19±1.05 | 25.59±1.07 |
| | GAT | 34.47±1.21 | 26.67±1.05 | 25.31±1.03 | 26.67±1.03 | 26.71±1.17 | 26.71±1.00 |
| Citeseer | GCN | 42.56±2.82 | 40.76±0.76 | 40.94±0.93 | 40.71±0.78 | 41.57±1.06 | 40.63±0.74 |
| | GAT | 47.42±3.26 | 40.62±0.90 | 40.44±0.62 | 40.55±0.90 | 40.99±0.97 | 40.61±0.95 |
| Pubmed | GCN | 21.31±0.71 | 20.20±0.36 | 20.24±0.38 | 20.20±0.36 | 20.05±0.41 | 20.17±0.36 |
| | GAT | 25.33±0.56 | 22.68±0.41 | 22.64±0.41 | 22.68±0.41 | 22.58±0.43 | 22.67±0.41 |
| Computers | GCN | 18.57±0.80 | 18.50±0.68 | 18.42±0.64 | 18.51±0.68 | 18.13±0.70 | 18.42±0.65 |
| | GAT | 16.79±0.80 | 16.76±0.75 | 16.61±0.64 | 16.76±0.75 | 16.84±0.73 | 16.75±0.73 |
| Photo | GCN | 11.72±0.66 | 11.60±0.59 | 11.62±0.65 | 11.60±0.60 | 11.67±0.51 | 11.56±0.56 |
| | GAT | 11.52±0.88 | 11.45±0.77 | 11.35±0.69 | 11.45±0.77 | 11.59±0.70 | 11.46±0.76 |
| CS | GCN | 10.28±0.27 | 10.16±0.20 | 10.20±0.19 | 10.16±0.20 | 10.60±0.42 | 10.14±0.21 |
| | GAT | 11.36±0.58 | 11.27±0.35 | 11.25±0.30 | 11.27±0.35 | 11.65±0.39 | 11.27±0.34 |
| Physics | GCN | 6.13±0.20 | 6.13±0.19 | 6.13±0.19 | 6.13±0.19 | 6.25±0.22 | 6.12±0.19 |
| | GAT | 6.54±0.18 | 6.54±0.18 | 6.53±0.17 | 6.54±0.18 | 6.64±0.18 | 6.53±0.18 |
| CoraFull | GCN | 52.32±0.68 | 52.09±0.51 | 52.01±0.49 | 52.08±0.51 | 52.20±1.20 | 51.61±0.54 |
| | GAT | 51.73±0.77 | 51.60±0.60 | 51.55±0.59 | 51.59±0.60 | 52.53±1.71 | 51.54±0.59 |

- **Calibration using spline (Spline)** [5] fits the calibration function with splines;

- **Dirichlet calibration (DIR)** [10] uses the Dirichlet distribution to model the distribution of probabilistic outputs. It also employs an off-diagonal and intercept regularization (ODIR);

- **Order invariant calibration (OI)** [17] is the order-invariant intra order-preserving model. It uses sorted output logits as calibration input and builds up a neural network with special structures to preserve the accuracy and the intra order of the predicted logits.

The authors of spline calibration specify how to calibrate a specific class or a chosen top-$r$ class, and in their implementation[4] they focus on calibrating the top-1 class. However, it is not clear how to adjust the rest of the predictions to ensure valid probabilistic predictions after calibration. We adopt a heuristic which proportionally rescales the non top-1 output probabilities so that the calibrated probabilistic output sums up to one. Also, the authors wrongly claimed that calibrating the top-1 score "does not alter the classification accuracy" [5]. In practice, the score after calibration might no longer remain top-1 and the predictions could be altered.

For Dirichlet calibration, we find out that the scaling factors $\lambda, \mu$ of ODIR affect the performance and need to be tuned depending on the dataset. Thus we do a hyperparameter search for each dataset with search space $(0.01, 0.1, 1, 10, 100, 1000, 10000, 100000)$.

Table 7 summarizes the calibration results in terms of ECE. Note that we do not include the results for Dirichlet calibration on CoraFull because it fails to calibrate the GNN backbones and significantly deteriorates the predictive accuracies ($< 10\%$ v.s. $> 60\%$ before calibration).

Overall we observe that GATS still achieves the state-of-the-art performance compared to the additional baselines for GNN calibration.

## C.3 Accuracies of calibration methods

Since many baseline calibration methods are not accuracy-preserving, in Table 8 we additionally report their test accuracies. Accuracy preserving methods (GATS, TS, ETS, CaGCN, OI) have the same accuracies as the uncalibrated case, which is also reported for reference.

---

[4]`https://github.com/kartikgupta-at-anu/spline-calibration`

Table 7: GNN calibration results (ECE, in percentage) with additional baselines.

| Dataset | Model | Uncal | IRM | Spline | DIR | OI | GATS |
|---|---|---|---|---|---|---|---|
| Cora | GCN | 13.04±5.22 | 3.69±1.17 | 4.89±1.27 | 3.93±1.26 | 4.83±1.50 | **3.64±1.34** |
| | GAT | 23.31±1.81 | 3.45±0.91 | 4.71±1.76 | 3.42±0.72 | 4.24±1.39 | **3.18±0.90** |
| Citeseer | GCN | 10.66±5.92 | 5.08±1.34 | 6.70±1.42 | 5.40±1.52 | 6.36±1.48 | **4.43±1.30** |
| | GAT | 22.88±3.53 | 4.15±1.50 | 6.07±1.77 | 4.87±1.36 | 6.08±1.30 | **3.86±1.56** |
| Pubmed | GCN | 7.18±1.51 | 1.64±0.58 | 1.72±0.46 | 1.42±0.33 | 1.23±0.44 | **0.98±0.30** |
| | GAT | 12.32±0.80 | 1.63±0.60 | 1.69±0.60 | **0.93±0.26** | 1.36±0.47 | 1.03±0.32 |
| Computers | GCN | 3.00±0.80 | 1.98±0.48 | **1.56±0.44** | 3.31±0.63 | 1.86±0.55 | 2.23±0.49 |
| | GAT | 1.88±0.82 | **1.32±0.35** | 1.56±0.53 | 2.23±0.73 | 2.17±0.72 | 1.39±0.39 |
| Photo | GCN | 2.24±1.03 | 1.53±0.47 | 1.68±0.57 | 1.61±0.60 | 1.75±0.49 | **1.51±0.52** |
| | GAT | 2.02±1.11 | 1.53±0.51 | 1.59±0.66 | 1.39±0.62 | 1.85±0.66 | **1.48±0.61** |
| CS | GCN | 1.65±0.92 | 1.29±0.32 | 1.08±0.38 | 0.90±0.19 | 1.55±0.50 | **0.88±0.30** |
| | GAT | 1.40±1.25 | 1.09±0.35 | 1.16±0.39 | 0.96±0.39 | 1.80±0.80 | **0.81±0.30** |
| Physics | GCN | 0.52±0.29 | 0.59±0.17 | 0.54±0.23 | **0.44±0.15** | 0.64±0.29 | 0.46±0.16 |
| | GAT | 0.45±0.21 | 0.56±0.16 | 0.45±0.18 | **0.42±0.14** | 0.60±0.32 | **0.42±0.14** |
| CoraFull | GCN | 6.50±1.26 | 4.33±0.77 | **2.92±0.79** | N/A | 10.61±1.40 | 3.76±0.74 |
| | GAT | 4.73±1.39 | 3.18±0.56 | **2.68±0.89** | N/A | 8.33±2.18 | 3.54±0.63 |

Table 8: Test accuracies of uncalibrated results (identical to those from accuracy-preserving methods) and calibrated predictions from non accuracy-preserving methods.

| Dataset | Model | Uncal | VS | IRM | Spline | DIR |
|---|---|---|---|---|---|---|
| Cora | GCN | 82.78±0.79 | 82.90±0.89 | 82.56±0.87 | 82.78±0.80 | 83.16±0.87 |
| | GAT | 81.98±0.92 | 82.98±0.77 | 81.74±1.04 | 81.98±0.92 | 82.77±0.85 |
| Citeseer | GCN | 72.19±0.82 | 72.06±0.90 | 72.04±0.79 | 72.16±0.82 | 72.31±0.99 |
| | GAT | 72.37±0.68 | 72.25±0.64 | 72.18±0.73 | 72.34±0.73 | 72.53±0.59 |
| Pubmed | GCN | 86.40±0.27 | 86.39±0.29 | 86.31±0.29 | 86.39±0.26 | 86.43±0.25 |
| | GAT | 84.46±0.34 | 84.55±0.38 | 84.25±0.38 | 84.44±0.34 | 84.62±0.35 |
| Computers | GCN | 88.13±0.56 | 88.19±0.56 | 88.20±0.54 | 88.12±0.55 | 87.78±0.65 |
| | GAT | 89.05±0.60 | 89.16±0.52 | 89.03±0.60 | 89.04±0.60 | 89.00±0.63 |
| Photo | GCN | 92.65±0.38 | 92.69±0.43 | 92.61±0.42 | 92.65±0.38 | 92.69±0.49 |
| | GAT | 92.65±0.54 | 92.76±0.45 | 92.58±0.57 | 92.64±0.54 | 92.92±0.44 |
| CS | GCN | 93.33±0.15 | 93.29±0.15 | 93.29±0.16 | 93.32±0.15 | 93.33±0.15 |
| | GAT | 92.57±0.25 | 92.57±0.22 | 92.54±0.24 | 92.56±0.24 | 92.60±0.22 |
| Physics | GCN | 95.99±0.14 | 95.98±0.14 | 95.98±0.15 | 95.98±0.14 | 96.00±0.14 |
| | GAT | 95.70±0.13 | 95.71±0.12 | 95.67±0.14 | 95.69±0.14 | 95.72±0.11 |
| CoraFull | GCN | 63.07±0.50 | 63.24±0.45 | 62.94±0.48 | 63.07±0.50 | N/A |
| | GAT | 63.00±0.59 | 63.10±0.52 | 62.85±0.59 | 63.00±0.58 | N/A |

## D Data efficiency and expressiveness of GATS: GAT results

Figure 7 shows the results for the GAT case. We see that GATS is also data efficient and expressive when calibrating GAT models.

## E CaGCN results discussion

While the ECEs of CaGCN in its original paper are promising [23], we observe that the ECEs of CaGCN are often unstable and sometimes even worse than that of the uncalibrated model in our experiments. One possible reason is that we use a different splitting from the CaGCN paper, where

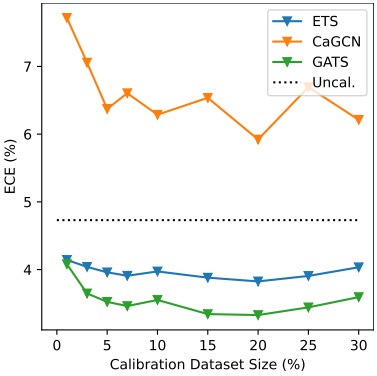

Figure 7: ECEs (in percentage) on CoraFull with GAT backbone for ETS, CaGCN, and GATS using various amounts of calibration data. Again, We observe that GATS is data-efficient and expressive. Here CaGCN fails to calibrate the GAT model on CoraFull.

they follow a fixed splitting from Kipf and Welling [7]. A significant difference is that the splitting from Kipf and Welling [7] has more validation nodes than the training nodes. This differs from typical real-world applications, where the larger fold would often be used to train a good classifier [13], and only the smaller fold is available for fitting the calibrator.

In our splitting, the validation sets of Cora and Citeseer are substantially smaller than those in Kipf and Welling [7]. We observe that CaGCN yields suboptimal calibration results (see Section 6.1) and predictions with higher confidence tend to be over-confident in the reliability diagram in Figure 8. By contrast, the validation set in Pubmed is relatively large since it has more nodes. We notice that CaGCN achieves competitive results in Pubmed and the confidence-accuracy curve almost lies on the diagonal. We observe that CaGCN also produces suboptimal calibration results in CoraFull, even though the validation set is large. We suspect that this is caused by the class imbalance of the CoraFull data. Class imbalance is known to be a challenge for many calibration methods [21].

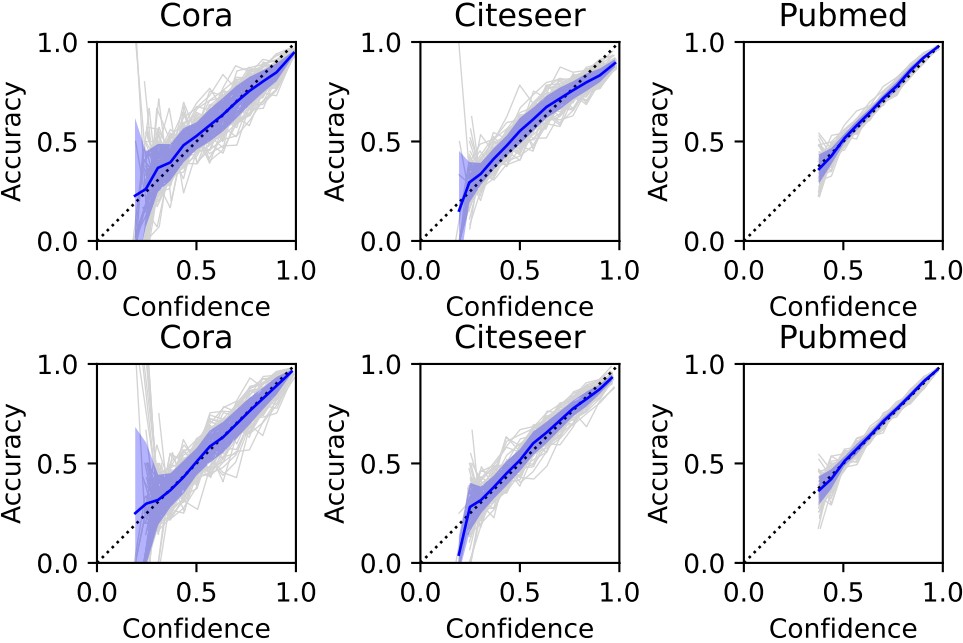

Figure 8: Top row: Reliability diagram of CaGCN for GCN trained on Cora, Citeseer, and Pubmed. Botom row: Reliability diagram of GATS for GCN trained on Cora, Citeseer, and Pubmed as a reference. We observe that CaGCN is noticeably over-confident in the high confidence region when the size of the validation set is relatively small (e.g., Cora and Citeseer).

# F  GATS weight visualization

GATS learns $\tau_i^h$ from sorted logits. We discover that the absolute value of the learned weights in the linear layer generally follows the ranking of the logits across the class. That is to say, logits with higher value have stronger influence to $\tau_i^h$. In Figure 9 we visualize the weights $\theta = (\theta^1, \ldots, \theta^H)$ of the linear layers $\phi^h$ in GATS. Here, it is interesting to see that the weights $\theta^h$ from different heads $h$ have slight variations. Combining multiple heads in the attention with sorting could be considered as a form of ensemble without the model being overly parameterized.

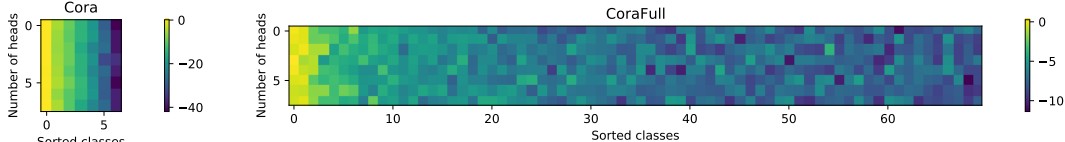

Figure 9: Visualization of the GATS weight $\theta$ on Cora (7 classes) and CoraFull (70 classes).

# G  Analysis of correlations between the factors

In this section we visualize the correlation between the local-view factors: the distance to training nodes, the relative confidence level, and the neighborhood similarity. Each plot shows how the factor on the y-axis varies when the factor on the x-axis is fixed to a given value. Two factors are independent when we observe a horizontal line in the plot. As the relative confidence level is a model dependent factor, GCN and GAT will have different correlation plots when it is involved.

## G.1  Distance to training nodes – relative confidence level

Figures 10, 11, 12, and 13 show the correlation plots between the distance to training nodes and the relative confidence level. In Figure 10 and 12 we see that regardless of the distance to the training nodes, the averaged relative confidence level stays around zero.

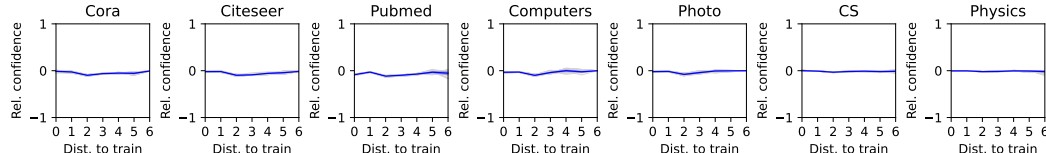

Figure 10: Relative confidence level of GCN results depending on the minimum distance to training nodes.

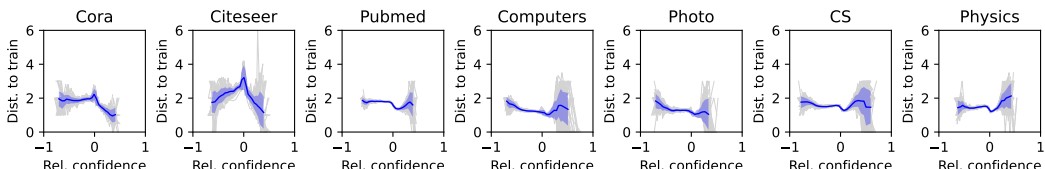

Figure 11: Minimum distance to training nodes depending on the relative confidence level of GCN predictions.

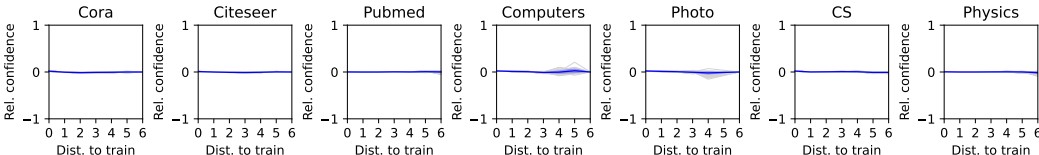

Figure 12: Relative confidence level of GAT results depending on the minimum distance to training nodes.

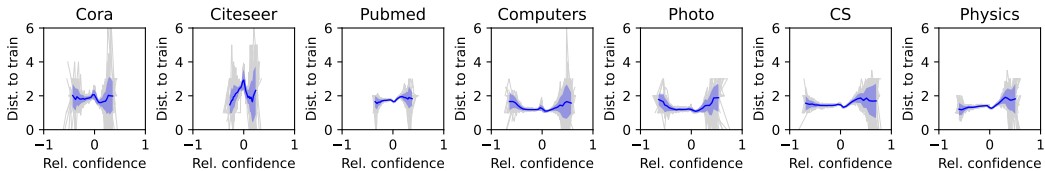

Figure 13: Minimum distance to training nodes depending on the relative confidence level of GAT predictions.

## G.2 Relative confidence level – neighborhood similarity

Figures 14, 15, 16, and 17 show the correlations between the relative confidence level and the neighborhood similarity. We observe some partial correlation between these two factors, especially in the negative region of the node homophily.

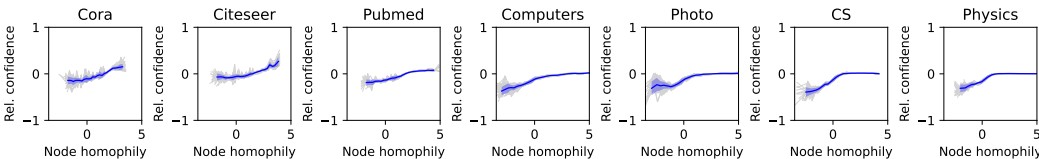

Figure 14: Relative confidence level of GCN predictions depending on the node homophily.

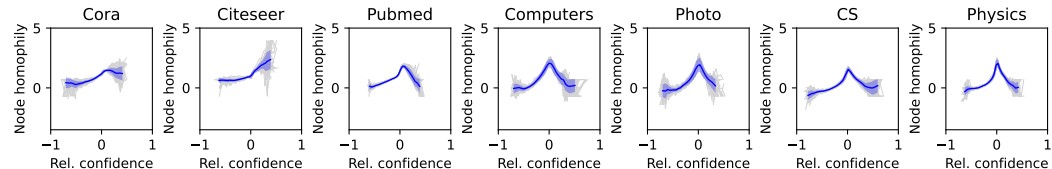

Figure 15: Node homophily depending on the relative confidence level of GCN predictions.

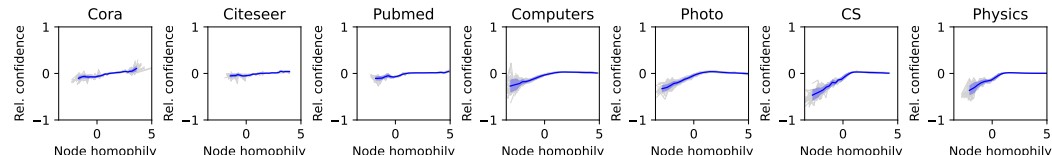

Figure 16: Relative confidence level of GAT predictions depending on the node homophily.

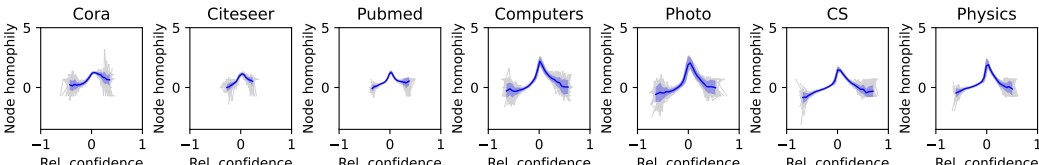

Figure 17: Node homophily depending on the relative confidence level of GAT predictions.

### G.3 Distance to training nodes – neighborhood similarity

Figures 18 and 19 show the correlation between the distance to training nodes and neighborhood similarity. Note that these two factors are not model-dependent and thus GCN and GAT share the same results. We observe that these two factors have a less significant correlation.

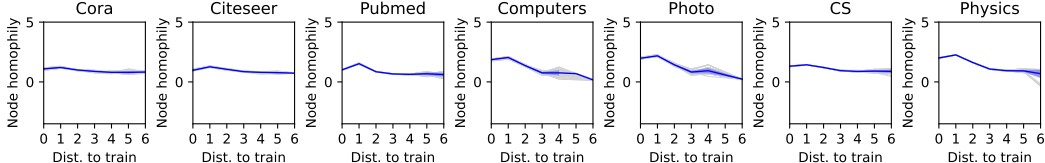

Figure 18: Node homophily depending on the minimum distance to training nodes.

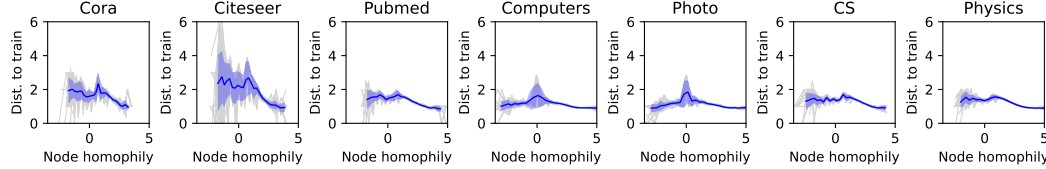

Figure 19: Minimum distance to training nodes depending on the node homophily.

## H   Factor node count analysis

In this section we plot the number of test nodes depending on the three local view factors: distance to training nodes, relative confidence level, and neighborhood similarity.

### H.1   Node count for distance to training nodes

Figure 20 summarizes the node count results of the distance to training nodes. We see the majority of nodes can be connected to the training nodes by one or two hops.

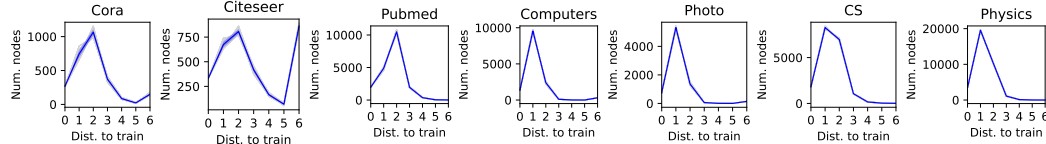

Figure 20: Number of nodes for the minimum distance to training nodes.

### H.2   Node count for relative confidence level

Figure 21 and 22 are the node count results of the relative confidence level for GCN and GAT respectively. We observe that most of the nodes are concentrated around the zero relative confidence level.

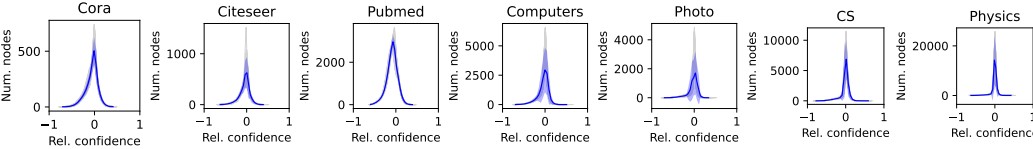

Figure 21: Number of nodes for the relative confidence level of GCN predictions.

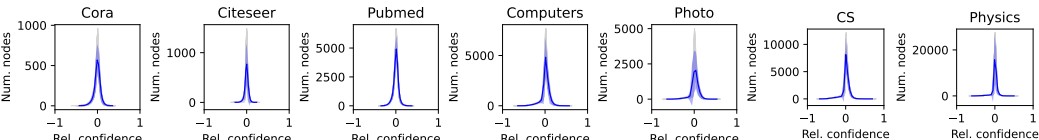

Figure 22: Number of nodes for the relative confidence level of GAT predictions.

### H.3 Node count for neighborhood similarity

Figure 23 shows that the majority of nodes lie in the positive homophily region.

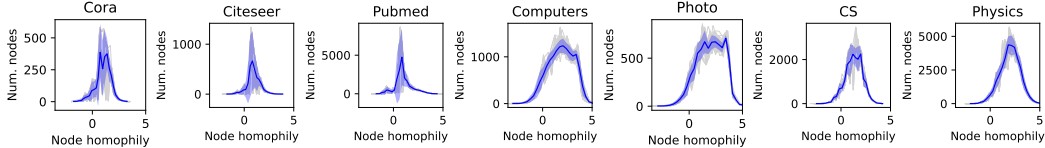

Figure 23: Number of nodes for the node homophily.