# OpenReview forum: "What Makes Graph Neural Networks Miscalibrated?"
_NeurIPS.cc/2022/Conference — NeurIPS 2022 Accept_

### Official Review · Reviewer_hxXe · 2022-07-07

**Rating:** 5
**Confidence:** 3
**Soundness:** 2 fair
**Presentation:** 3 good
**Contribution:** 3 good

**Summary:**

This work studies prediction calibration for GNNs on node prediction. It performs empirical analysis to identify five factors that are correlated with the GNN prediction confidence. Then an attention-based temperature scaling method is proposed to calibrate GNNs. Experiments show that the proposed method achieves lower ECE compared to baselines.


**Questions:**

In figure 1, several plots show that the low confidence predictions achieve super high accuracy close to 1. Why is this the case?


**Strengths And Weaknesses:**

Strengths:

Studying GNN calibration by analyzing its specific challenges that are different from CNNs is a good contribution.
The result in terms of ECE seems promising.
The paper presentation is mostly clear.

Weaknesses:

I have some concerns about the five factors proposed by the paper.

The definition and analysis for some of the five factors are a bit vague. For example, Section 3.2 is regarding the "diversity of node distributions" factor, which is analyzed through the "entropy distribution" plots in Figure 2. If I understand them correctly, in Figure 2, the entropy of each GNN output is computed, and then a density estimation based on all such entropy values is plotted. In this case, the name "node distribution" can be confusing because the factor is actually the distribution of GNN predictions on nodes. Node distribution should refer to the actual node features/labels, and it is a property of the data. Also, "diverse entropy implies uniform temperature scaling is insufficient" doesn't seem quite right. We can have a very large uniform temperature to make all predictions very uniform or a very small uniform temperature to make all predictions close to deterministic. Both cases will make the entropy distribution very concentrated. The latter can generate a plot close to the GNN case. Furthermore, the plots in the second row seem quite similar to me, which questions whether this factor is always necessary because it is dataset-dependent.

The analysis of these five factors is not thorough given some of these factors may be correlated. For example, the relative confidence level and the neighborhood similarity should be correlated. Intuitively, we are unlikely to have neighbors with the same prediction but very different confidence, or different predictions and both very high confidence. Then doing a more detailed analysis can help to better understand their correlations and importance and maybe disentangle the factors. In the current version, although the ablation study in Table 2 suggests all of the factors help with some ECE reduction, their significants are actually questionable. Like for $\delta c_i$, removing it only increases the ECE by less than 0.1% whereas the std is around 1%.

As a core contribution of the paper, what factors truly influence the GNN prediction and confidence is the key contribution. Ideally, we would want fewer factors and more disentangled factors. I feel the factor definition and analysis can be further improved.

---

> ### Author Response · Authors · 2022-08-02
> **Author Response to Reviewer hxXe**
>
> Thank you for your valuable feedback!
>
> To answer your concerns regarding the factors that influence the GNN calibration:
>
> > [...] the name "node distribution" can be confusing because the factor is actually the distribution of GNN predictions on nodes. Node distribution should refer to the actual node features/labels, and it is a property of the data.
>
> We are sorry for the confusion caused by the term "node distributions". By node distributions we mean the nodewise predictive distributions, which have diverse confidence levels as shown by the entropy histograms. In particular, we first compute the entropy of each GNN node prediction, and then visualize the entropy distribution. We have rephrased this term in the paper revision in order to make it clearer.
>
> > Also, "diverse entropy implies uniform temperature scaling is insufficient" doesn't seem quite right. We can have a very large uniform temperature to make all predictions very uniform or a very small uniform temperature to make all predictions close to deterministic. Both cases will make the entropy distribution very concentrated. The latter can generate a plot close to the GNN case.
>
> We agree that uniform temperature scaling can recalibrate the globally over/under-confident tendency. However, in general, temperature scaling cannot fully fix individual (per sample) miscalibration errors unless all predictions have the same confidence level, which would be the case for concentrated entropy histograms. Section 3.2 shows that the entropy histograms are less concentrated for GNN predictions compared to the image classification baseline, which shows that uniform temperature scaling might be insufficient to fully tackle the GNN calibration problem.
>
> > Furthermore, the plots in the second row seem quite similar to me, which questions whether this factor is always necessary because it is dataset-dependent.
>
> The entropy histograms GNN predictions on Photo, CS, and Physics are still visibly less concentrated than the image classification case.  This suggests that GNN predictions are more complex and diverse, and that nodewise adaptive calibration can be beneficial.
>
> > The analysis of these five factors is not thorough given some of these factors may be correlated. For example, the relative confidence level and the neighborhood similarity should be correlated. [...] Then doing a more detailed analysis can help to better understand their correlations and importance and maybe disentangle the factors.
>
> These five factors are summarized from empirical observations and each of them describes a distinct aspect of GNN calibration. Our goal is to lay out these factors for a better understanding of GNN calibration. While some might be partially correlated, there is no direct causal relationship between these factors. We follow your suggestion and have plotted additional figures to analyze the correlations among the three nodewise factors in Appendix G of the revised supplementary. We indeed notice some partial correlations, and your intuition is reflected in the correlation plots (Figures 14 and 16) for many datasets. Thank you for the suggestion!
>
> >  In the current version, although the ablation study in Table 2 suggests all of the factors help with some ECE reduction, their significants are actually questionable. Like for $\delta c_i$, removing it only increases the ECE by less than 0.1% whereas the std is around 1%.
>
> Compared to other factors, the performance increase of the relative confidence level $\delta c_i$ is indeed less significant. We speculate that this may be related to its partial correlation with the factor neighborhood similarity, which may make up for its effect on improving the ECE. The results of GNNs are known to have a high variance that is hard to reduce even with repeated runs. For each experiment, we do a total of 75 runs with various initial weights and random splits to ensure the reliability of the results.
>
> > In figure 1, several plots show that the low confidence predictions achieve super high accuracy close to 1. Why is this the case?
>
> This is an artifact of the reliability diagram: when there are only few prediction samples inside a bin (e.g., in a low confidence region), the estimated accuracy of that bin could be inaccurate and have high variance. This is also indicated by the large confidence intervals.

---

> > ### Comment · Reviewer_hxXe · 2022-08-05
> > **Response to Authors**
> >
> > Thank the authors for the response and for putting effort into adding new experiment results, especially given the tight rebuttal time window. I think the correlation plots help readers better understand these factors. My specific questions are mostly answered. My general question about what should be the most essential and disentangled factors, however, remains. Like the authors said, the current facotrs are empirical observations, some of them are correlated, and some of them have much smaller impact compared to others. My question is for sure a hard question to answer given the tight rebuttal time, and probably beyond the scope of this paper. In any case, there should be room for improvement. Maybe new observations can be made and another new factor appears to be a substitute of several current factors, etc. I encourage the authors to try to address it in future works.

---

### Official Review · Reviewer_rsXN · 2022-07-10

**Rating:** 8
**Confidence:** 4
**Soundness:** 4 excellent
**Presentation:** 4 excellent
**Contribution:** 4 excellent

**Summary:**

This paper considers confidence calibration for GNN node classification. It conducted a very throughout and careful empirical study of the factors in graph structure that may effect the calibration. It discovered 5 factors that contributing to miscalibration: general under-confidence, diversity of distribution, distance to training, relative confidence level, and neighborhood similarity. After demonstrating each factor independently, the authors proposed a temperature scaling method where each node is assigned with a different temperature, based on a formula that is inspired by the empirical analysis. The proposed GATS method is parameteric efficient but also captures the effects that influences confidence calibration. In particular, GATS out performs other calibration methods for most datasets.

**Questions:**

N/A

**Strengths And Weaknesses:**

Strength
  - the GNN calibration problem is important but underexplored
  - the empirical study is conducted very carefully, the results are very inspiring
  - the proposed method is very simple, interpretable and effective
  - the ablation study is very informative
  - the logic flow and writing is very clear

Weakness
  - the authors should explain how the GATS parameters are learned
  - the author could compare GATS & GaGCN in terms of the number of parameters

---

> ### Author Response · Authors · 2022-08-02
> **Author Response to Reviewer rsXN**
>
> Thank you for the positive feedback! We are really glad that you find the paper to be well-written and technically sound.
>
> To answer your specific questions:
>
> > the authors should explain how the GATS parameters are learned
>
> GATS is trained on the validation set using the negative log-likelihood (NLL) loss and validated on the training set for early stopping and hyperparameter search.
> We use Adam as the optimizer and initialize $T_0$, $\gamma_t$, and $\gamma_n$ to 1 and $\omega$ to 0.
>
> Details about the model training and calibration setups are provided in Sections A.2 and A.3 in the supplementary material. We invite you to check them out. Also, we have provided our source code which reproduces the training and calibration processes in practice.
>
> > the author could compare GATS & CaGCN in terms of the number of parameters
>
> In Section 4.2 we mention that GATS has $C \cdot H + 4$ parameters ($ 8 C + 4$ in practice as we use 8 heads). CaGCN uses another two-layer GCN model on top of the base model for calibration. According to [1], the hidden dimension of the GCN model is 16 units. Thus, the first layer and the second layer consist of $C \cdot 16 + 16$ and $16 \cdot 1 + 1$ respectively. In total, the CaGCN uses $16 C + 33$ parameters which are more than double the parameters of GATS.
> Besides, we want to point out that the learning process on GATS is relatively easy as we use the sorted logits as input.
> A single head GATS with parameters $C+4$ can already achieve competitive results as shown in Table 3.
>
> [1] X. Wang, H. Liu, C. Shi, and C. Yang. Be confident! towards trustworthy graph neural networks via confidence calibration. In NeurIPS, 2021.

---

### Official Review · Reviewer_mbVJ · 2022-07-10

**Rating:** 6
**Confidence:** 4
**Soundness:** 3 good
**Presentation:** 4 excellent
**Contribution:** 3 good

**Summary:**

Calibration of neural networks is an important area of research. This paper provides empirical insights on the miscalibration of GNNs for node classification, and then proposes an approach that takes all those factors into account to calibrate the GNN outputs.

**Questions:**

Q1: The "distance to training nodes" seems to lead to miscalibration mostly on Cora and Citeseer, and not on the the other graphs. This is also true about the "general under-confident tendency". Is there a reason for this? Is there a correlation between the two factors? Are Cora and Citeseer different in any specific way that makes them more miscalibrated for nodes far away from the training set?
More evidence may be necessary to back up these two factors.

Q2: From what I understand, "diversity of node distributions" is a direct effect of "general under-confident tendency". That is, if the model is under confident in its probability for the correct class, it means other classes have higher probabilities, and so the entropy is high. Is there anything novel about 3.2 then?

Q3: Among the five factors, I was mostly surprised/impressed by "relative confidence level". It seems like a good finding and the analysis in Fig 3 shows the importance of it. However, I was disappointed when I saw the ablation results; none of the degradations due to removing \delta c_i seem to be statistically significant. Is there a reason for the mismatch between what figure 3 shows regarding the importance of this factor, and what Table 2 shows regarding the unimportance of it? Is it because the proposed calibration algorithm doesn't utilize this factor in the best way?

Q4: Figs 1 and 3 seem to imply that the GCN outputs are well-calibrated for the physics dataset. However, Fig 4 implies that for a subset of the nodes (those with low relative confidence), the outputs are quite miscalibrated. Can you explain how these results can all hold simultaneously? Are there only very few nodes with low relative confidence for Physics (and that's why there is an extremely large std)?

Q5: This is more of an observation than a question, but "distance to training nodes" and "neighborhood similarity" are both factors that affect how "easy" the prediction is for a node, in the sense that, e.g., nodes with many neighbors of the same class are highly probable to belong to the same class (under homophily -- which the studied datasets all exhibit highly). Therefore, it is not quite surprising that these two factors play a role in miscalibration. Nevertheless, seeing empirical backup for the two is definitely nice.

**Limitations:**

1- Among the five factors studied in the paper, the first one has been previously studied, the second one seems to be a direct effect of the first one, the third-one and the fifth one are not quite surprising, and the fourth one is quite interesting but doesn't seem to have a significant effect according to the ablation studies.

2- According to the ablation study, many of the sub-components of the proposed algorithm seem to have negligible effects on the results for many of the datasets (although it is partly understandable because some components may make up for the others).

Nevertheless, I believe a dedicated work for studying multiple factors about the miscalibration of GNNs and providing a calibration approach based on those factors could be beneficial to the community.

**Strengths And Weaknesses:**

Strengths:
+ Well written
+ Good, in-depth analysis of different factors
+ Tested on a suite of 7/8 datasets
Weaknesses:
- Some of the findings from the analyses in section 3 do not seem to match with the ablation study results.
- Some of the identified factors are somewhat expected, and not surprising

---

> ### Author Response · Authors · 2022-08-02
> **Author Response to Reviewer mbVJ**
>
> We sincerely thank the reviewer's constructive feedback.
>
> We address your questions and concerns as follows:
>
> __Response to Q1:__
>
> Cora and Citeseer tend to have different characteristics compared to other datasets which might lead to more pronounced trends for some factors. We have included the statistics of the datasets we considered in Appendix A.1 of the supplementary, which show that they have noticeably smaller graph sizes and sparser connections. Other than this, the distance to the training nodes is a nodewise feature, whereas the general under-confident tendency is a global trend, and there is no direct relation between the two factors.
>
> __Response to Q2:__
>
> "General under-confident tendency" means that GNN predictions tend to be less confident than they should be. That is, the predicted confidence of GNN is lower than the averaged accuracy. However, the predictions do not necessarily have low confidence on an absolute scale. "Diversity of node distributions" means the nodewise predictions from GNN have varying confidence levels, not uniformly low confidence predictions (i.e. more dispersed instead of concentrated high entropy values). Hence, "diversity of node distributions" and "general under-confident tendency" are distinct.
>
> __Response to Q3:__
>
> The relatively insignificant contribution of the "relative confidence level" to the GATS calibration results might be related to its correlation with neighborhood similarity, as shown in Appendix G of the revised supplementary. Thus the attention mechanism might partially cover its role in improving GNN calibration. There might also be alternative ways to incorporate this factor in GNN calibration methods, which will be interesting to investigate in future work.
>
> __Response to Q4:__
>
> The uncalibrated GCN outputs are well calibrated for the physics dataset as indicated by the low ECE in Table 1. Indeed, there are only very few nodes with low relative confidence for Physics. As shown in Appendix H of the revised supplementary, nodes tend to have near-zero relative confidence levels and the miscalibration in low relative confidence has little effect over the averaged ECE.
>
> __Response to Q5:__
>
> We agree with the reviewer that these two important factors are related to how "easy" the prediction is for a node. One of the main goals of this work is to thoroughly study the factors that may affect miscalibration. We point them out as they are both important factors that should be considered for GNN calibration.
>
> __Response to L1:__
>
> These five factors are summarized from empirical observations, and each factor describes a distinct aspect of GNN calibration. Note that "diversity of node distributions" is not a direct effect of "general under-confident tendency" (see response to Q2 above). While some factors might not be so surprising, they are all important and should be considered when designing GNN calibration methods.
>
> __Response to L2:__
>
> If we understand correctly from Q3, you are mainly referring to the relative confidence level $\delta c_i$ which indeed provides a minor improvement, especially compared to other factors. The results of GNNs are known to have a high variance that is hard to reduce even with repeated runs. For each experiment, we run our model 75 times with different initial weights and random splits to ensure the reliability of the results.

---

### Official Review · Reviewer_hr2j · 2022-07-22

**Rating:** 6
**Confidence:** 2
**Soundness:** 3 good
**Presentation:** 2 fair
**Contribution:** 3 good

**Summary:**

The paper proposed a new method to solve the GNN calibration problem. It identifies several factors that make the calibration different on GNNs from general multi-class classification tasks. Based on the factors a new calibration method is developed which produces nodewise temperature scaling using an attention-based architecture. Experiments demonstrate that the proposed GATS is accurate, data-efficient and expressive.

**Questions:**

see above weakness questions.

**Limitations:**

Yes.

**Strengths And Weaknesses:**


Strengths:

* A thorough study of the factors which impact the calibration error of GNNs
* A new temperature scaling model which considers the factors and mitigates the calibration problem in GNNs

Weakness:

* The writing is generally clear, but as a reader who is not familiar with calibration, it is not easy to understand how the calibration error is calculated and why the GCN model has a different pattern from general multi-class classification. I think it is better to explain it a little bit in the beginning of the method.

* Section 3 and Section 4 should be aligned better. It seems all factors in Sec 3 are used later, but not all of them are directly corresponding a parameter in Eq. (5) (e.g. h in Eq. (2)). Since there are too many new brough-in parameters in Eq(5), it should be explained more. Some notations can be reclaimed (\eta_{c_i}), some notations should be described more clearly (e.g. what does the h mean in \tau_i^h ).

---

> ### Author Response · Authors · 2022-08-02
> **Author Response to Reviewer hr2j**
>
> Thank you for your feedback! To address your concerns:
>
> > The writing is generally clear, but as a reader who is not familiar with calibration, it is not easy to understand how the calibration error is calculated and why the GCN model has a different pattern from general multi-class classification. I think it is better to explain it a little bit in the beginning of the method.
>
> We have included the formulas for ECE in Appendix A.3 of the supplementary material and provided a reference [1] for ECE computation and neural network calibration in Section 5. Following your suggestion, we have added a pointer to Appendix A.3 in the revised paper where ECE is first mentioned.
> We will make use of the additional page from the camera-ready version to include more related background.
>
> [1] C. Guo, G. Pleiss, Y. Sun, and K. Q. Weinberger. On calibration of modern neural networks. ICML, 2017.
>
> > Section 3 and Section 4 should be aligned better. It seems all factors in Sec 3 are used later, but not all of them are directly corresponding a parameter in Eq. (5) (e.g. h in Eq. (2)). Since there are too many new brough-in parameters in Eq(5), it should be explained more. Some notations can be reclaimed (\eta_{c_i}), some notations should be described more clearly (e.g. what does the h mean in \tau_i^h ).
>
> The node homophily in $h$ denotes the node homophily that is computed in Figure 5. We have removed this notation to avoid further confusion.
> $\tau_i^h$ is the nodewise temperature contribution for $h$-th attention head since we use a multi-head formulation.
> In Section 4.1 paragraph "Formulation of $T_i$", we have a list that explains how the five factors are taken into account in our GATS method. This list aligns the factors in Section 3 to the different parts of GATS design in Section 4 which, when put together, results in Equation 5. We rephrase some descriptions in Section 4.1 and establish more connections between Section 3 and Section 4.1.

---

> > ### Comment · Reviewer_hr2j · 2022-08-09
> > **Response to authors**
> >
> > Thank the authors for their clarification. Most of my questions have been answered. I still think the paper is interesting and have good contributions, but I also agree with other reviewers that it has limitations in the novelty of some factors and the correlation of the factors. I will keep my score.

---

### Meta-Review · Area_Chair_sMYU · 2022-08-24

**Recommendation:** Accept
**Confidence:** Less certain

**Metareview:**

This paper studies the calibration problem for GNN node classification. It identifies 5 factors contributing to miscalibration: general under-confidence, diversity of distribution, distance to training, relative confidence level, and neighborhood similarity. A temperature scaling method is proposed where each node is assigned with a different temperature. All reviewers vote for accept. However, multiple reviewers have raised concerns on the validity of the 5 factors. I encourage the authors to thoroughly address them in the revised version.

**Award:**

No

---

### Decision · Program_Chairs · 2022-09-14

Accept